# DBA: Efficient Transformer with Dynamic Bilinear Low-Rank Attention

## Abstract

Many studies have been conducted to improve the efficiency of Transformer from quadric to linear over long sequence conditions. Among them, the low-rank-based methods aim to learn the projection matrices to compress sequence length, thus achieving efficiency gain. However, the projection matrices are fixed once they have been learned, which compress sequence length with dedicated coefficients for tokens in the same position regardless of different sequences. Adopting such input-invariant low-rank projections ignores the fact that the most informative part of a sequence varies from sequence to sequence, thus failing to preserve the most useful information that lies in varied positions of different sequences. In addition, previous efficient Transformers only focus on the influence of sequence length while neglecting the effect of hidden state dimension to achieve further efficiency gain. To address the aforementioned problems, we present an efficient yet effective attention mechanism, namely **Dynamic Bilinear Low-Rank Attention (DBA)**, which compresses sequence length by input-sensitive dynamic projection matrices and achieves linear time and space complexity by jointly optimizing sequence length and hidden state dimension while maintaining state-of-the-art performance. Specifically, we first theoretically demonstrate that the sequence length can be compressed losslessly from a novel perspective of information theory, with the compression matrices dynamically determined by the input sequence. Furthermore, we show that the hidden state dimension can be approximated by extending the Johnson–Lindenstrauss lemma and achieves high-order small amount error, optimizing the attention in bilinear form. In addition, theoretical analysis shows that DBA is proficient in capturing high-order relations in cross-attention problems. Experiments over tasks with diverse sequence length conditions show that DBA achieves state-of-the-art performance compared with various strong baselines while maintaining less memory consumption with higher speed, demonstrating the effectiveness and efficiency of DBA.

## 1 Introduction

The Transformer (Vaswani et al., 2017) has shown immense capabilities in a wide range of areas, including natural language processing (Dai et al., 2019), computer vision (Dosovitskiy et al., 2021; Liu et al., 2021), time series analysis (Zerveas et al., 2021), and multi-modal tasks (Qin et al., 2022a; Yu et al., 2019). However, the Vanilla Transformer suffers quadratic time and memory complexity, raising concerns about its further application scenarios. Therefore, several efficient Transformers have been introduced (Tay et al., 2022). Among them, kernel-based methods have drawn much attention due to their optimization-friendly characteristic, which improves the efficiency by using the approximation in the attention mechanism (Katharopoulos et al., 2020; Wang et al., 2020; Ma et al., 2021; Xiong et al., 2021; Qin et al., 2022b; Choromanski et al., 2021). One popular kernel-based technique is low-rank approximation, which compresses sequence length dimension using the same coefficients for all sequences. For instance, Wang *et al.* (Wang et al., 2020) approximated the stochastic matrix in sequence length dimension by using sets of fixed coefficients learned in the training process to calculate the weighted sum of tokens in different positions. Xiong *et al.* (Xiong et al., 2021) adopted the Nyström method to approximate the attention mechanism to linear complexity, decreasing the sequence length with mean pooling.

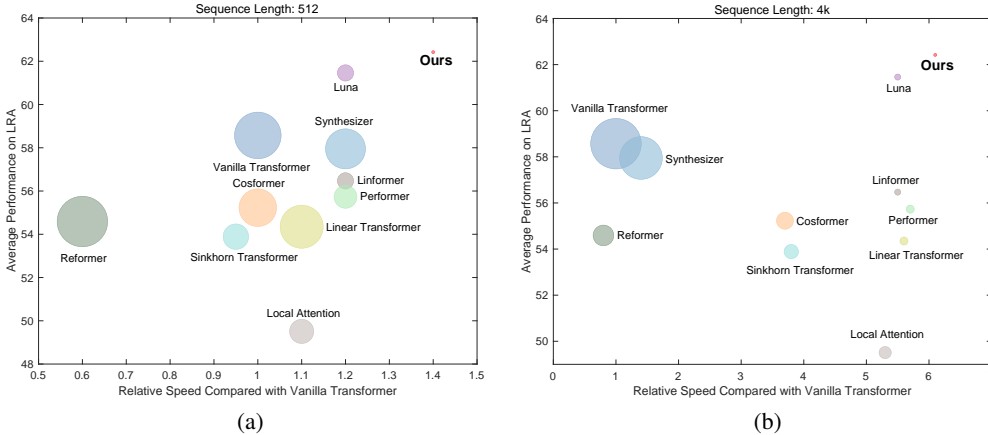

Figure 1: Performance (y-axis, higher is better), speed (x-axis, higher is better), and memory footprint (circle sizes, smaller is better) of efficient Transformers on the Long-Range Arena benchmark (Tay et al., 2021b) compared with Vanilla Transformer (Vaswani et al., 2017) in different sequence length conditions (512 and 4k). **DBA could achieve state-of-the-art performance with the highest speed and lowest memory consumption over various sequence length conditions.**

However, the flexibility of low-rank projection in previous methods is limited. The projection matrices are pre-determined or fixed after the training process, which compress different sequences by using the same coefficients for tokens in the same position. Such input-invariant low-rank compressions ignore the fact that the informative part of a sequence varies from sequence to sequence. Hence, the compression might fail to preserve the most informative parts lying in different positions and limit the performance over tasks where the most informative parts of inputs change significantly, such as image-related tasks. In addition, previous efficient Transformers only focused on optimizing the sequence length while ignoring the influence of hidden state dimension. The hidden state dimension also contributes to the computation cost and becomes more critical to efficiency when processing moderate or short sequences. Previous efficient Transformers that achieve significant memory compression and speed-up rate in long sequence conditions could end up with similar efficiency when processing moderate or short sequences compared with the Vanilla Transformer, as shown in Figure 1.

To address the aforementioned problems, we proposed an efficient yet effective attention mechanism, namely Dynamic Bilinear Low-Rank Attention (DBA), which compresses sequence length with input-sensitive dynamic projection matrices and achieves linear computation and memory efficiency with bilinear optimization from both sequence length and hidden state dimension. Specifically, we first theoretically show that sequence length can be compressed losslessly from a novel perspective of the information theory, where the projection matrices are dynamically determined by the input sequence to best preserve the most informative parts. Furthermore, we demonstrate that the hidden state dimension can be approximated by extending the Johnson–Lindenstrauss lemma (Arriaga & Vempala, 2006; Lindenstrauss & Johnson, 1984) with high-order small amount error. In addition, theoretical analysis shows that DBA is able to capture high-order relations in cross-attention problems, which is crucial to the performance in multi-modality tasks.

Extensive experiments over tasks with various sequence length conditions are conducted on three different datasets, including Long-Range Arena (LRA) (Tay et al., 2021b) as the long sequence benchmark, UEA multivariate time series classification archive (Bagnall et al., 2018) to evaluate the performance of various sequence lengths, VQA-v2 (Goyal et al., 2017) as the illustrations of DBA in capturing high-order relations. The DBA achieves state-of-the-art performance with impressing speed-up and memory compression rate compared with other competitors over various sequence length conditions, demonstrating the effectiveness and efficiency of DBA in a wide range of applications.

Our main contributions can be summarized as follows:

1) We introduce an efficient and effective attention mechanism, namely Dynamic Bilinear Low-Rank Attention (DBA), which compresses sequence length by input-sensitive dynamic projection matrices. The DBA achieves efficiency gain over various sequence length conditions with linear time and space complexity by jointly optimizing sequence length and hidden state dimension.

2) Theoretical guarantee from information theory and matrix low-rank approximation demonstrates that DBA has a similar capability to the Vanilla Attention with low expected error. In addition, theoretical analysis shows that DBA is able to capture high-order inter-relations in cross-attention problems.

3) Extensive experiments on tasks with various sequence length conditions show that DBA could achieve state-of-the-art performance in a wide range of applications with impressing efficiency gain. In addition, DBA is superior in capturing high-order relations in the cross-attention task, which outperforms the Vanilla Transformer based MCAN (Yu et al., 2019) with only 12% of parameters in the attention layer.

## 2 BACKGROUND AND RELATED WORK

### 2.1 VANILLA TRANSFORMER

The Vanilla Transformer (Vaswani et al., 2017) uses Vanilla Attention as its main algorithm, which is calculated via the softmax weighted sum of all values $V$ concerning weights obtained by the multiplication of $Q$ and $K$:

$$P_\phi(K, Q) = \text{softmax}\left(\frac{QK^T}{\sqrt{d}}\right) \tag{1}$$

$$\text{Attention}\left(P_\phi(K, Q), V\right) = P_\phi(K, Q)V \tag{2}$$

Here we define $P_\phi(K, Q)$ as attention map, and later, we will abbreviate $P_\phi(K, Q)$ as $P_\phi$ for simplicity. The $Q \in \mathbb{R}^{n \times d}$, $K \in \mathbb{R}^{n \times d}$, $V \in \mathbb{R}^{n \times d}$, and $P_\phi \in \mathbb{R}^{n \times n}$, where $n$ is sequence length and $d$ indicates hidden state dimension. Notice that the time and memory complexity for the Vanilla Attention is proportional to $O\left(n^2 d\right)$. For long sequence applications, the impact of $n$ becomes dominant, and the influence of $d$ becomes greater when facing moderate or short sequences.

### 2.2 EFFICIENT TRANSFORMERS

One kind of Efficient Transformers is by using sparsity, where each token could only access limited perspective fields with fixed or learned patterns, including the local attention (Parmar et al., 2018), Reformer (Kitaev et al., 2020), Sinkhorn (Tay et al., 2020), Routing Transformer (Roy et al., 2021), ALiBi (Press et al., 2022), Learning-to-Hash Attention (LHA) (Sun et al., 2022), YOSO (Zeng et al., 2021), ClusterFormer (Wang et al.), Poolingformer (Zhang et al., 2021), and Big Bird (Zaheer et al., 2020). To make the attention have wider perspective fields, some works concentrate on the interactions between near field and far field, such as Focal Attention (Yang et al., 2021), FMMformers (Nguyen et al., 2021), Long-Short Transformer (Zhu et al., 2021), and Crossformer (Wang et al., 2022).

Another popular approach is kernel-based method, which improves the efficiency of Transformer by rewriting the multiplication in equations 1 and 2, such as Linear Transformer (Parmar et al., 2018), PoNet (Tan et al., 2022), Random Feature Attention (Peng et al., 2021), and LARA (Zheng et al., 2022). In (Choromanski et al., 2021; Qin et al., 2022b; Chen et al., 2021b), the authors optimize the softmax kernel with faster reweighting functions. Since the kernels are the approximation in attention matrices, they can also be optimized by low-rank methods, such as Linformer (Wang et al., 2020), Luna (Ma et al., 2021), and Nyströmformer (Xiong et al., 2021). In (Zhuang et al., 2022; Luo et al., 2021; Kreuzer et al., 2021; Zhou et al., 2022), the authors improve the efficiency of attention kernel by exploring the frequency domain. Some works also focus on the multi-head characteristic in attention mechanism and optimize via reducing the parallel computations, such as FLASH (Hua et al., 2022) and Transformer-MGK (Nguyen et al., 2022).

The proposed DBA is most similar to Linformer, which both approximate features in Vanilla Attention to the low-rank matrices and achieve linear complexity. The main differences are in four

folds. First, the low-rank projection matrices in DBA are more flexible than in Linformer, which are dynamically determined by the input sequence rather than fixed after training to best preserve the most informative parts of a sequence. Secondly, DBA could process sequences in various lengths as the dimensions of sequence length compression matrices are also determined by the input. Thirdly, DBA could achieve state-of-the-art performance with high efficiency over various sequence length conditions due to jointly considering the sequence length and hidden state dimension. Furthermore, DBA is proficient in capturing high-order relations with multi-stage interactions, whereas Linformer could only perform one-stage interaction.

## 3 METHOD

Our goal is to design an efficient attention mechanism with linear complexity, where the analysis starts with the Vanilla Attention defined in equations 1 and 2. In Section 3.1, we will theoretically demonstrate that the input sequence length $n$ can be compressed losslessly from the perspective of information theory, leading to linear complexity in both time and space. In Section 3.2, we will extend the Johnson–Lindenstrauss lemma to prove that the multiplication between $\boldsymbol{Q}$ and $\boldsymbol{K}$ can be reduced by low-rank approximation with high-order small amount error in the results, mitigating the impact of hidden state dimension $d$ on efficiency. In Section 3.3, we will present the source of matrices newly introduced to DBA and show that the sequence length compression matrices are dynamically determined by the input features, leading to adaptive coefficients for tokens in the same position. In Section 3.4, we will show that DBA could capture high-order inter-relations in cross-attention problems and perform multi-stage interactions within a single attention layer.

### 3.1 OPTIMIZE THE SEQUENCE LENGTH WITH INFORMATION THEORY

In this section, we will optimize the quadric complexity of sequence length to Transformer by analyzing the attention mechanism from the information theory perspective, leading to linear complexity in both time and space. Specifically, we will show that the attention matrix $\boldsymbol{P}_\phi \in \mathbb{R}^{n \times n}$ could be replaced by a set of smaller matrices without information loss.

Note that in the Vanilla Attention, $\boldsymbol{P}_\phi$ is deterministic for dedicated $\boldsymbol{QK}^T$. Hence, we could derive that the conditional entropy between $\boldsymbol{QK}^T$ and $\boldsymbol{P}_\phi$ is 0.

$$H(\boldsymbol{P}_\phi | \boldsymbol{QK}^T) = H(\text{softmax}\left(\frac{\boldsymbol{QK}^T}{\sqrt{d}}\right) | \boldsymbol{QK}^T) = 0 \tag{3}$$

Therefore, $\boldsymbol{QK}^T$ contains all the information $\boldsymbol{P}_\phi$ has. Notice that $\boldsymbol{QK}^T$ could be reconstructed losslessly with the based of $\boldsymbol{QK}^T$ and the reconstruction coefficients. Hence, the conditional entropy between the bases of $\boldsymbol{QK}^T$ with reconstruction coefficients to the $\boldsymbol{P}_\phi$ is 0.

$$H(\boldsymbol{P}_\phi | \text{ basis}_r(\text{basis}_c(\boldsymbol{QK}^T)), \boldsymbol{W}_r', \boldsymbol{W}_c') = 0 \tag{4}$$

where $\text{basis}_r$ and $\text{basis}_c$ calculate the basis of $\boldsymbol{QK}^T$ in the row and column spaces, respectively. $\boldsymbol{W}_r'$ and $\boldsymbol{W}_c'$ are the reconstruction coefficients for row and column, which values and dimensions are determined by $\boldsymbol{QK}^T$.

From the properties of matrix rank in multiplication, we could get the following inequality.

$$\text{Rank}\left(\boldsymbol{QK}^T\right) \leq \max\left(\text{Rank}\left(\boldsymbol{Q}\right), \text{Rank}\left(\boldsymbol{K}\right)\right) \leq \min\left(n, d\right) \tag{5}$$

where $\text{Rank}()$ calculates the rank of a matrix.

Hence, the dimension of $\text{basis}_r(\text{basis}_c(\boldsymbol{QK}^T))$ are no larger than $\mathbb{R}^{\min(n,d) \times \min(n,d)}$. Therefore, with the help of equation 4, a given $\boldsymbol{P}_\phi$ can be represented losslessly with a matrix $\boldsymbol{P}_\phi' \in \mathbb{R}^{d_p \times d_p}$ ($d_p \leq \min\left(n, d\right)$) and reconstruction coefficient $\boldsymbol{W}_r' \in \mathbb{R}^{n \times d_p}$ and $\boldsymbol{W}_c' \in \mathbb{R}^{n \times d_p}$. In practice, we could form:

$$\boldsymbol{P}_\phi = \boldsymbol{W}_r' \boldsymbol{P}_\phi' \boldsymbol{W}_c'^T \tag{6}$$

where $\boldsymbol{P}_\phi'$, $\boldsymbol{W}_r'$, and $\boldsymbol{W}_c'$ are determined by the input and learned through the training process.

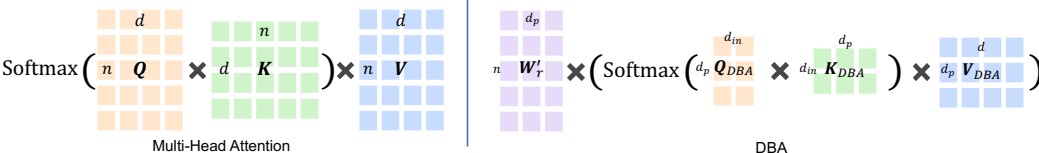

Figure 2: Illustration of the algorithm of Vanilla Attention versus DBA. Compared with Vanilla Transformer (Vaswani et al., 2017), the $\boldsymbol{Q}, \boldsymbol{K}$ in DBA are compressed to low-rank alternatives with bilinear form in both sequence length and the hidden state dimension. The DBA has the same input and output as Vanilla Transformer by using reconstruction matrix $\boldsymbol{W}_r'$, which easily makes DBA plug and play with existing Transformers. The labels on the row and column of squares represent the dimension of features.

Here, we could project $\boldsymbol{Q} \in \mathbb{R}^{n \times d}, \boldsymbol{K} \in \mathbb{R}^{n \times d}$ to $\boldsymbol{Q}_l \in \mathbb{R}^{d_p \times d}, \boldsymbol{K}_l \in \mathbb{R}^{d_p \times d}$ to generate the $\boldsymbol{P}_\phi'$, with $\boldsymbol{Q}_l$ with $\boldsymbol{K}_l$ as the input for equation 1. Therefore, we could derive the following equation:

$$
\begin{aligned}
\boldsymbol{P}_\phi = \boldsymbol{W}_r' \boldsymbol{P}_\phi' \boldsymbol{W}_c'^{T} &= \boldsymbol{W}_r' \left( \text{softmax} \left( \frac{\boldsymbol{Q}_l \boldsymbol{K}_l^T}{\sqrt{d}} \right) \right) \boldsymbol{W}_c'^{T} \\
&= \boldsymbol{W}_r' \left( \text{softmax} \left( \frac{(\boldsymbol{W}_r \boldsymbol{Q})(\boldsymbol{K}^T \boldsymbol{W}_c^T)}{\sqrt{d}} \right) \right) \boldsymbol{W}_c'^{T}
\end{aligned}
\tag{7}
$$

where $\boldsymbol{W}_r \in \mathbb{R}^{d_p \times n}, \boldsymbol{W}_c \in \mathbb{R}^{d_p \times n}, \boldsymbol{W}_r' \in \mathbb{R}^{n \times d_p}$, and $\boldsymbol{W}_c' \in \mathbb{R}^{n \times d_p}$.

By proposing $\boldsymbol{P}_\phi'$ as the new attention map instead of $\boldsymbol{P}_\phi$, we avoid quadric complexity in time and space in attention map generation. However, notice that the reconstruction process using $\boldsymbol{W}_r'$ and $\boldsymbol{W}_c'^{T}$ still brings high complexity. Here, we first merge the $\boldsymbol{W}_c'$ and $\boldsymbol{V}$ first as $\boldsymbol{V}_{DBA}$, then the $\boldsymbol{V}_{DBA}$ is multiplied by $\boldsymbol{P}_\phi'$. The reconstruction process by $\boldsymbol{W}_r'$ is set in the last. By optimizing the calculation order, the DBA achieves linear complexity.

$$
\text{DBA}(\boldsymbol{K}, \boldsymbol{Q}, \boldsymbol{V}) = \boldsymbol{W}_r'(\boldsymbol{P}_\phi'(\boldsymbol{W}_c'^{T} \boldsymbol{V})) = \boldsymbol{W}_r'\left(\boldsymbol{P}_\phi' \boldsymbol{V}_{DBA}\right)
\tag{8}
$$

where $\boldsymbol{V}_{DBA} = \boldsymbol{W}_c'^{T} \boldsymbol{V} \in \mathbb{R}^{d_p \times d}$.

## 3.2 Optimize Hidden State Dimension with Matrix Approximation

In Section 3.1, we optimize sequence length by analyzing from the information theory perspective, leading to linear complexity. In this section, we will further increase the efficiency of DBA by mitigating the impact of hidden state dimension $d$ on efficiency. Specifically, we extend the Johnson–Lindenstrauss lemma (Arriaga & Vempala, 2006; Lindenstrauss & Johnson, 1984) to show the multiplication between $\boldsymbol{Q}$ and $\boldsymbol{K}$ can be approximated with high-order small amount error. Based on the Johnson–Lindenstrauss lemma, we could derive that when $d_{in} \geq 10 log(d_p) / (\epsilon^2 - \epsilon^3)$, the following equation holds.

$$
\Pr\left(||(\boldsymbol{W}_r \boldsymbol{Q}) \boldsymbol{R} \boldsymbol{R}^T \left(\boldsymbol{K}^T \boldsymbol{W}_c^T\right) - (\boldsymbol{W}_r \boldsymbol{Q}) \left(\boldsymbol{K}^T \boldsymbol{W}_c^T\right)|| \leq \epsilon ||(\boldsymbol{W}_r \boldsymbol{Q}) \left(\boldsymbol{K}^T \boldsymbol{W}_c^T\right)||\right) > 1 - o(1)
\tag{9}
$$

The proof details are in Appendix A.1.

The equation 9 shows that the multiplication between $\boldsymbol{Q}$ and $\boldsymbol{K}$ could be replaced by alternatives with lower hidden state dimension ($d$ vs. $d_{in}$) and achieves errors in high-order small quantities compared to full-rank multiplication. Therefore, we could further project $\boldsymbol{Q}_l \in \mathbb{R}^{d_p \times d}, \boldsymbol{K}_l \in \mathbb{R}^{d_p \times d}$ mentioned in Section 3.1 to $\boldsymbol{Q}_{DBA} \in \mathbb{R}^{d_p \times d_{in}}, \boldsymbol{K}_{DBA} \in \mathbb{R}^{d_p \times d_{in}}$, and finally, the DBA could be written as follows:

$$
\begin{aligned}
\text{DBA}(\boldsymbol{K}, \boldsymbol{Q}, \boldsymbol{V}) &= \boldsymbol{W}_r' \left( \left( \text{softmax} \left( \frac{((\boldsymbol{W}_r \boldsymbol{Q})\boldsymbol{R})(\boldsymbol{R}^T(\boldsymbol{K}^T \boldsymbol{W}_c^T))}{\sqrt{d_{in}}} \right) \right) \left( \boldsymbol{W}_c'^{T} \boldsymbol{V} \right) \right) \\
&= \boldsymbol{W}_r' \left( \text{softmax} \left( \frac{\boldsymbol{Q}_{DBA} \boldsymbol{K}_{DBA}}{\sqrt{d_{in}}} \right) \boldsymbol{V}_{DBA} \right)
\end{aligned}
\tag{10}
$$

The attention mechanism is now compressed with bilinear form in both sequence length and the hidden state dimension, increasing the efficiency for sequences with various lengths. The graphical comparison between Vanilla Attention and DBA is illustrated in Figure 2.

### 3.3 THE SOURCE OF MATRICES

In this section, we will define the source of matrices that are newly introduced to DBA in self-attention situation, including the hidden state compression matrix $R$, the sequence compression matrices $W_r$, $W_c$, and the reconstruction matrices $W_r'$, $W_c'$. We will show that the weights in sequence compression matrices are determined by the input sequence, leading to dynamic coefficients for tokens in the same position between different sequences.

The $R$ compress hidden state with a fixed dimension. Therefore, it is set as a fully connected layer and learned through training. The $W_r'$, $W_c'$ are set as the input sequences propagate thought fully connected layers to obtain the expected hidden state dimension. The $W_r$ and $W_c$ are generated by combining the input sequence and an extra input $Z \in \mathbb{R}^{d_p \times d}$ in a shorter length.

$$W_r = \varphi\left(ZQ^T\right) \tag{11}$$

$$W_c = \varphi(ZK^T) \tag{12}$$

where $\varphi$ is a normalization function to stabilize the training process. In practice, we set $\varphi$ as softmax function.

Therefore, the compression matrices $W_r$ and $W_c$ are dynamically determined by the input sequence, where every coefficient in $W_r$ and $W_c$ is the linear transformations of token features in the corresponding position. Each row in $W_r$ and $W_c$ is a set of compression coefficients for all tokens in the input sequence, and the results of each position in the final compressed sequences $W_r Q$ and $K^T W_c^T$ are the different weighted sum of tokens in the original sequence. The weights in the reconstruction matrices $W_r'$, $W_c'$ are also determined by the input, where the rows also represent different sets of coefficients dynamically determined by the input sequence. Note that the dimensions for both $W_r$, $W_c$ and $W_r'$, $W_c'$ are dynamically determined by the input, making them able to process sequences in various lengths without fixed padding. In practice, we set $Z$ as learnable parameters propagating through different attention layers.

### 3.4 CAPTURE HIGH-ORDER RELATIONS IN CROSS-ATTENTION

In this section, we will show that DBA is able to capture high-order relations with multi-stage interactions within an attention layer in the cross-attention situation. We will first introduce the cross-attention algorithm in Vanilla Transformer and then compare it with the proposed DBA.

The cross-attention in Vanilla Transformer shares the same expression as self-attention in equations 1 and 2. The only difference is input. In cross-attention, one input $X_1$ from $H_1$ is processed as $Q_1$, and the other input $X_2$ from $H_2$ is processed as $K_2$, $V_2$, where the subscript 1, 2 indicate the variables in different hierarchies, and $H_1$, $H_2$ denote different hierarchies. By leveraging the Vanilla Attention algorithm, $Q_1$ is fused with $K_2$, leading to one-stage interaction within an attention layer.

$$\text{Cross-Attention}\left(K_2, Q_1, V_2\right) = \text{softmax}\left(\frac{Q_1 K_2^T}{\sqrt{d}}\right) V_2 \tag{13}$$

The DBA takes different inputs compared with Vanilla Attention in cross-attention. Instead of taking the full-length features $X_2$ as $K_2$, $V_2$, DBA takes the compressed sequence $W_{r2} X_2$ as $Z_1$, $K_2$, and $V_2$. Both models take $X_1$ as $Q_1$.

Firstly, we compress sequence length following Section 3.1. As $W_{r2} X_2$ formed $K_2$, $V_2$ are already been compressed, we only need to compress sequence length in $Q_1$ using $W_{r1}$, which is obtained from $Z_1$.

$$W_{r1} = \varphi(Z_1 Q_1^T) = \varphi(\text{Linear}(W_{r2} X_2) Q_1^T) \tag{14}$$

where Linear() denotes fully connected layer.

The advantages of using $W_{r2} X_2$ as $Z_1$ are in two folds. First, the features from two different hierarchies interact when generating $W_{r1}$. Second, it compresses the sequence length of $Q_1$, where the compression coefficients are guided by features in $H_2$.

Table 1: Speed and peak memory consumption of different models on byte-level text classification with various sequence lengths (256, 512, 1k, 2k, 3k, and 4k). The average performances on the LRA task are listed on the right. The best model is made bold.

| Model | Speed ↑ | | | | | | Peak Memory Usage ↓ | | | | | | Avg. ↑ |
|---|---|---|---|---|---|---|---|---|---|---|---|---|---|
| | 256 | 512 | 1k | 2k | 3k | 4k | 256 | 512 | 1k | 2k | 3k | 4k | |
| Vanilla Transformer | 1.0 | 1.0 | 1.0 | 1.0 | 1.0 | 1.0 | 1.0 | 1.0 | 1.0 | 1.0 | 1.0 | 1.0 | 58.57 |
| Local Attention | 1.0 | 1.1 | 1.1 | 1.7 | 3.2 | 5.3 | 0.94 | 0.75 | 0.49 | 0.29 | 0.19 | 0.14 | 49.51 |
| Linformer | 1.0 | 1.2 | 1.2 | 1.9 | 3.7 | 5.5 | 0.90 | 0.70 | 0.44 | 0.21 | 0.18 | 0.10 | 56.47 |
| Reformer | 1.0 | 0.6 | 0.5 | 0.4 | 0.7 | 0.8 | 1.21 | 1.06 | 0.70 | 0.37 | 0.28 | 0.24 | 54.59 |
| Sinkhorn Trans. | 0.9 | 1.0 | 1.1 | 1.6 | 2.9 | 3.8 | 0.92 | 0.76 | 0.55 | 0.31 | 0.21 | 0.16 | 53.89 |
| Synthesizer | **1.2** | 1.2 | 1.1 | 1.2 | 2.9 | 1.4 | 1.06 | 0.91 | 0.76 | 0.75 | 0.74 | 0.74 | 57.95 |
| Cosformer | 0.9 | 1.0 | 1.0 | 1.5 | 2.8 | 3.7 | 1.24 | 0.88 | 0.58 | 0.28 | 0.25 | 0.19 | 55.23 |
| Linear Transformer | 1.0 | 1.1 | 1.1 | 1.9 | 3.7 | 5.6 | 1.16 | 0.95 | 0.44 | 0.22 | 0.15 | 0.11 | 54.35 |
| Performer | 1.1 | 1.2 | 1.2 | 1.9 | 3.8 | 5.7 | 0.89 | 0.74 | 0.44 | 0.22 | 0.15 | 0.11 | 55.73 |
| Luna | 1.0 | 1.2 | 1.2 | 1.8 | 3.7 | 5.5 | 0.88 | 0.70 | 0.44 | 0.23 | 0.17 | 0.10 | 61.46 |
| **DBA** | 1.1 | **1.4** | **1.4** | **2.0** | **4.1** | **6.1** | **0.84** | **0.66** | **0.38** | **0.19** | **0.15** | **0.09** | **62.21±0.21** |

Table 2: Performance on the LRA benchmark. The DBA is trained with 5 random seeds, and the average scores with accuracy variances are reported. The best model is made bold.

| Model | ListOps ↑ | Text ↑ | Retrieval ↑ | Image ↑ | Pathfinder ↑ | Avg. ↑ |
|---|---|---|---|---|---|---|
| Vanilla Transformer (Vaswani et al., 2017) | 36.37 | 64.27 | 78.38 | 42.44 | 71.40 | 58.57 |
| Local Attention (Parmar et al., 2018) | 15.82 | 52.98 | 70.65 | 41.46 | 66.63 | 49.51 |
| Sparse Transformer (Child et al., 2019) | 17.07 | 63.58 | 72.53 | 44.24 | 71.71 | 53.83 |
| Sinkhorn (Tay et al., 2020) | 33.67 | 61.20 | 65.88 | 41.23 | 67.45 | 53.89 |
| Linear Transformer (Katharopoulos et al., 2020) | 16.13 | 65.90 | 72.09 | 42.34 | 75.30 | 54.35 |
| Reformer (Kitaev et al., 2020) | 37.27 | 56.10 | 73.03 | 38.07 | 68.50 | 54.59 |
| cosformer (Qin et al., 2022b) | 37.90 | 63.41 | 61.36 | 43.17 | 70.33 | 55.23 |
| Fnet (Lee-Thorp et al., 2022) | 35.33 | 65.11 | 59.61 | 38.67 | 77.80 | 55.30 |
| Performer (Choromanski et al., 2021) | 18.01 | 65.40 | 75.43 | 42.77 | 77.05 | 55.73 |
| Longformer (Beltagy et al., 2020) | 35.63 | 62.85 | 68.32 | 42.22 | 69.71 | 55.75 |
| Linformer (Wang et al., 2020) | 35.70 | 53.94 | 77.83 | 38.56 | 76.34 | 56.47 |
| Synthesizer (Tay et al., 2021a) | 36.99 | 61.68 | 80.04 | 41.61 | 69.45 | 57.95 |
| Big Bird (Zaheer et al., 2020) | 36.05 | 64.02 | 76.41 | 40.83 | 74.87 | 58.44 |
| Nyströmformer (Xiong et al., 2021) | 37.15 | 65.52 | 79.56 | 41.58 | 70.94 | 58.95 |
| YOSO (Zeng et al., 2021) | 37.40 | 64.28 | 77.61 | 44.67 | 71.86 | 59.16 |
| Scatterbrain (Chen et al., 2021a) | **38.60** | 64.55 | 80.22 | 43.65 | 69.91 | 59.39 |
| Pixelfly (Chen et al., 2022) | 37.65 | **66.78** | 80.55 | 42.35 | 72.01 | 59.87 |
| Luna (Ma et al., 2021) | 37.43 | 65.74 | 79.38 | 46.39 | 78.36 | 61.46 |
| **DBA** | 38.10±0.40 | 66.25±0.04 | **80.64±0.01** | **46.51±0.50** | **79.56±0.10** | **62.21±0.21** |

After sequence length compression, we optimize the impact of $d$ on efficiency following Section 3.2, and finally, we could get $(\boldsymbol{Q}_{DBA})_1$, $(\boldsymbol{K}_{DBA})_2$, and $(\boldsymbol{V}_{DBA})_2$ to perform second interaction between two features as in equation 10. Therefore, DBA could capture inter-relations in the sequence compression matrices generation procedure and attention mechanism within a single attention layer, where the compressed feature in $H_2$ interacts with original and compressed features in $H_1$, making DBA able to capture high-order relations and perform multi-stage interactions.

## 4 EXPERIMENTS

We evaluate the performance of DBA on three datasets, covering long and diverse sequence conditions with self- and cross-attention tasks, including Long-Range Arena (LRA) (Tay et al., 2021b) as the benchmark on long sequence, UEA multivariate time series classification archive (Bagnall et al., 2018) to evaluate performance on various sequence lengths, VQA-v2 (Goyal et al., 2017) to test the performance of cross-attention. The detailed descriptions of datasets are in Appendix A.2, and the experiment settings are listed in Appendix A.3.

Table 3: Performance on the UEA multivariate time series classification archive. The best model is made bold.

| Task / Model | Transformer | Linformer | Performer | Linear | Cosformer | Flowformer | DBA |
|---|---|---|---|---|---|---|---|
| Ethanolconcentration ↑ | 32.7 | 32.6 | 31.2 | 33.5 | 31.9 | 33.8 | **35.4±1.1** |
| Facedetection ↑ | 67.3 | 67.0 | 67.0 | 67.1 | 67.0 | 67.6 | **68.7±0.3** |
| Handwriting ↑ | 32.0 | 28.9 | 32.1 | 34.7 | 34.7 | 33.8 | **35.1±0.1** |
| Heartbeat ↑ | 76.1 | 76.1 | 78.0 | 75.6 | 76.6 | 77.6 | **78.0±1.0** |
| Japanese vowels ↑ | 98.7 | 98.6 | 98.1 | 99.2 | 99.2 | 98.9 | **99.6±0.1** |
| Pems-Sf ↑ | 82.1 | 82.3 | 80.9 | 80.9 | 82.1 | 83.8 | **84.1±0.3** |
| Selfregulationscp1 ↑ | 92.2 | 91.8 | 91.5 | 91.8 | 92.5 | 92.5 | **92.8±0.3** |
| Selfregulationscp2 ↑ | 53.9 | 57.2 | 56.7 | 55.6 | 56.7 | 56.1 | **58.1±0.3** |
| Spokenarabicdigits ↑ | 98.4 | 98.8 | 98.4 | 98.8 | 98.0 | 98.8 | **99.5±0.1** |
| Uwavegesturelibrary ↑ | 85.6 | 84.7 | 85.3 | 85.0 | 85.0 | 86.6 | **87.4±0.1** |
| Average Accuracy ↑ | 71.9 | 71.8 | 71.9 | 72.2 | 72.4 | 73.0 | **73.9±0.4** |

## 4.1 EFFICIENCY

The efficiency of DBA compared with Vanilla Transformer and other efficient Transformers are illustrated in Figure 1 and Table 1. We report the speed and peak memory usage of different attentions in 256-4k sequence lengths. The DBA achieves state-of-the-art efficiency in terms of speed and peak memory usage, which is faster than the Vanilla Transformer and consumes fewer memories over various sequence conditions. In the long sequence conditions, DBA is 6.1 times faster than Vanilla Transformer and only uses 9% of memory in 4k sequence length. As for shorter sequence length, DBA could also achieve the highest efficiency among others, with 1.4 times faster and only uses 66% of memories compared to the Vanilla Attention in 512 sequence length. The DBA only falls behind the Synthesizer when facing 256-sequence length in terms of speed. However, DBA uses much less memory with much-suppressed efficiency on the long sequence condition. In addition, DBA achieves the best among others in terms of average performance on the LRA task, demonstrating the effectiveness and efficiency of DBA.

## 4.2 PERFORMANCE ON LONG SEQUENCE MODELING

We evaluate long sequence modeling performance of DBA and the previous methods on the LRA benchmark, as listed in Table 2. The DBA achieves state-of-the-art performance in terms of average score. By closer observation of each individual task, DBA achieves the best results on three out of five individual tasks. Notably, DBA suppresses the Vanilla Transformer and previous low-rank-based methods in all five tasks and is exceptionally proficient in image-related tasks where the most informative parts change significantly for different inputs. Note that DBA has the fastest speed and lowest memory consumption in all tasks, demonstrating the effectiveness and efficiency of DBA.

## 4.3 PERFORMANCE ON TIME SERIES SIGNAL IN VARIOUS LENGTH

We use UEA multivariate time series classification archive to evaluate the performance of models in various sequence lengths. The results are illustrated in Table 3. The DBA achieves the best performance compared with previous methods in all 10 tasks, with 2.3% improvement on the average accuracy compared with the Vanilla Transformer, highlighting the capability of processing sequences in various lengths.

## 4.4 PERFORMANCE ON CAPTURE CROSS-ATTENTION RELATIONS

We use the VQA-v2 dataset to evaluate the performance of DBA in cross-attention tasks. The results are shown in Table 4. Compared with the previous methods, where the image and question interact once per layer, DBA could capture high-order relations between hierarchies and perform multi-stage interactions within an attention layer, making DBA achieve the best results on the VQA-v2 tasks in all four evaluation aspects with only 12% of parameters compared with Vanilla Transformer based

Table 4: Performance on the *val* split of VQA-v2 dataset. `nPara` denotes the number of parameters in attention layers. The best model is made bold.

| Model | nPara ↓ | All ↑ | Y.N. ↑ | Num. ↑ | Other ↑ |
|---|---|---|---|---|---|
| MCAN (Yu et al., 2019) | 44M | 67.17 | 84.82 | 49.37 | 58.48 |
| BAN (Kim et al., 2018) | 60M | 66.00 | 83.61 | 47.04 | 57.62 |
| Linformer (Wang et al., 2020) | 39M | 66.19 | 83.77 | 48.15 | 57.61 |
| Performer (Choromanski et al., 2021) | 38M | 65.64 | 83.27 | 46.83 | 57.21 |
| MCAoAN (Rahman et al., 2021) | 58M | 67.24 | 84.95 | 49.51 | 58.45 |
| **DBA** | **5.2M** | **68.53±0.01** | **85.60±0.07** | **50.60±0.17** | **60.30±0.02** |

Table 5: Efficiency and performance of DBA with state space model on LRA dataset.

| Task | S4 (Gu et al., 2022) | | | S4+DBA | | |
|---|---|---|---|---|---|---|
| | Accuracy ↑ | Speed ↑ | Memory↓ | Accuracy ↑ | Speed ↑ | Memory↓ |
| ListOps ↑ | 59.60 | 1.0 | 1.0 | 59.70 | 1.3 | 0.8 |
| Text ↑ | 86.20 | 1.0 | 1.0 | 85.40 | 1.2 | 0.9 |
| Retrieval ↑ | 90.90 | 1.0 | 1.0 | 91.39 | 1.4 | 0.9 |
| Image ↑ | 87.28 | 1.0 | 1.0 | 86.90 | 1.5 | 0.8 |
| Pathfinder ↑ | 94.20 | 1.0 | 1.0 | 93.98 | 1.4 | 0.8 |
| Avg. ↑ | 83.64 | 1.0 | 1.0 | 83.47 | 1.4 | 0.8 |

MCAN (Yu et al., 2019) in attention layer, highlighting the effectiveness of DBA in capturing cross-attention relations.

### 4.5 PERFORMANCE WITH STATE SPACE MODEL BACKBONE

As a different approach from Transformer, the state space model has achieved promising results in long sequence modeling. The state space model takes the similar input $X \in \mathbb{R}^{n \times d}$ as the Transformer when processing the sequence, with its speed and memory consumption much influenced by the sequence length $n$.

The DBA can also be directly plug-and-play to the state space model by compressing the sequence length the state space models need to process from $\mathbb{R}^{n \times d}$ to $\mathbb{R}^{d_p \times d}$ to improve the efficiency while maintaining the final performance. We use the S4 (Gu et al., 2022) as backbone. The results are illustrated in Table 5. The S4 with DBA optimization could achieve 1.4x average speed boost and 0.8x average memory consumption with competitive performance compared to the baseline, highlighting the universality of DBA.

## 5 CONCLUSION

In this paper, we propose Dynamic Bilinear Low-Rank Attention (DBA), an efficient attention mechanism that compresses sequence length by input-sensitive dynamic projection matrices and achieves linear time and space complexity by jointly optimizing sequence length and hidden state dimension. Theoretical analysis from the information theory and matrix low-rank approximation perspectives shows that DBA could achieve a similar function to Vanilla Attention with high-order small amount error. In addition, DBA is capable of capturing high-order relations in cross-attention problems. Experiments show that DBA is able to achieve state-of-the-art performance with faster speed and lower memory consumption compared with previous models, highlighting its efficiency and effectiveness.

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

# A APPENDIX

## A.1 PROOF IN OPTIMIZING HIDDEN STATE DIMENSION

***Johnson–Lindenstrauss lemma.*** *Let $\boldsymbol{R} \in \mathbb{R}^{d \times d_{in}}$, $1 \leq d_{in} \leq d$, with i.i.d. entries from $\mathcal{N}(0, 1/d)$. For any $\mathbf{x}, \mathbf{y} \in \mathbb{R}^d$, we have*

$$\Pr\left(\left|\left|\mathbf{x}\boldsymbol{R}\boldsymbol{R}^T\mathbf{y}^T - \mathbf{x}\mathbf{y}^T\right|\right| \leq \epsilon \left|\left|\mathbf{x}\mathbf{y}^T\right|\right|\right) > 1 - 2e^{-\left(\epsilon^2 - \epsilon^3\right)d_{in}/4} \tag{15}$$

Based on the Johnson–Lindenstrauss lemma, we could obtain:

$$\Pr\left(\left|\left|\boldsymbol{Q}\boldsymbol{R}\boldsymbol{R}^T\boldsymbol{k}_i{}^T - \boldsymbol{Q}\boldsymbol{k}_i{}^T\right|\right| \leq \epsilon \left|\left|\boldsymbol{Q}\boldsymbol{k}_i{}^T\right|\right|\right)$$
$$\geq 1 - \sum_{\boldsymbol{q}_i \in \boldsymbol{Q}} \Pr\left(\left|\left|\boldsymbol{q}_i\boldsymbol{R}\boldsymbol{R}^T\boldsymbol{k}_i{}^T - \boldsymbol{q}_i\boldsymbol{k}_i{}^T\right|\right| > \epsilon \left|\left|\boldsymbol{q}_i\boldsymbol{k}_i{}^T\right|\right|\right) > 1 - 2ne^{-\left(\epsilon^2 - \epsilon^3\right)d_{in}/4} \tag{16}$$

The $\boldsymbol{K}$ contain $n$ rows. Here we could get:

$$\Pr\left(\left|\left|\boldsymbol{Q}\boldsymbol{R}\boldsymbol{R}^T\boldsymbol{K}^T - \boldsymbol{Q}\boldsymbol{K}^T\right|\right| \leq \epsilon \left|\left|\boldsymbol{Q}\boldsymbol{K}^T\right|\right|\right)$$
$$\geq 1 - \sum_{\boldsymbol{k}_i \in \boldsymbol{K}} \Pr\left(\left|\left|\boldsymbol{Q}\boldsymbol{R}\boldsymbol{R}^T\boldsymbol{k}_i{}^T - \boldsymbol{Q}\boldsymbol{k}_i{}^T\right|\right| > \epsilon \left|\left|\boldsymbol{Q}\boldsymbol{k}_i{}^T\right|\right|\right) > 1 - 2n^2 e^{-\left(\epsilon^2 - \epsilon^3\right)d_{in}/4} \tag{17}$$

Hence,

$$\Pr\left(\left|\left|\left(\boldsymbol{W}_r\boldsymbol{Q}\right)\boldsymbol{R}\boldsymbol{R}^T\left(\boldsymbol{K}^T\boldsymbol{W}_c^T\right) - \left(\boldsymbol{W}_r\boldsymbol{Q}\right)\left(\boldsymbol{K}^T\boldsymbol{W}_c^T\right)\right|\right| \leq \epsilon \left|\left|\left(\boldsymbol{W}_r\boldsymbol{Q}\right)\left(\boldsymbol{K}^T\boldsymbol{W}_c^T\right)\right|\right|\right)$$
$$> 1 - 2d_p{}^2 e^{-\left(\epsilon^2 - \epsilon^3\right)d_{in}/4} \tag{18}$$

Let $d_{in} \geq 10log\left(d_p\right)/\left(\epsilon^2 - \epsilon^3\right)$, we could derive the equation 9, then theorem follows.

## A.2 DATASETS

LRA (Tay et al., 2021b) is a popular benchmark to test the efficiency of Transformers in long sequence conditions, containing a suite of tasks (ListOps (Nangia & Bowman, 2018), byte-level text classification (Maas et al., 2011), document retrieval (Radev et al., 2013), pixel-level image classification (Krizhevsky & Hinton, 2009), and Pathfinder (Linsley et al., 2018)) with sequence length ranging from 1k to 4k.

UEA multivariate time series classification archive (Bagnall et al., 2018) is a collection of datasets to evaluate the time series classification algorithms, which contains a wide range of problems in various sequence length conditions.

VQA-v2 (Goyal et al., 2017) is a popular benchmark for multi-modal models, containing 1.1 million human-labeled image-question pairs with around 13 million associated answers on 200k images from the Microsoft COCO dataset (Lin et al., 2014), and it is split into the *train*, *val*, and *test* set.

Table 6: Summary of experiment benchmarks.

| Dataset | Task | Sequence Length |
|---------|------|-----------------|
| LRA (Tay et al., 2021b) | Long Sequence Modeling | 1k-4k |
| UEA (Bagnall et al., 2018) | Time Series | 29-1751 |
| VQA-v2 (Goyal et al., 2017) | Visual Question Answering | 5-625 |

## A.3 EXPERIMENT SETTINGS

All the experiments are conducted using PyTorch (Paszke et al., 2019) and Numpy (Harris et al., 2020) with Nvidia GPU.

For the experiment on the LRA dataset, DBA follows the configurations as (Ma et al., 2021), where all models use the same data processing strategy and model architecture for fair comparisons.

For the experiment on the UEA multivariate time series classification archive, we select 10 multi-variate datasets similar to (Wu et al., 2022) and use the same configurations following (Zerveas et al., 2021). As some of the tasks contain sequences of various lengths, we padded the batched input to the maximum length of the task during training process. During implementation, DBA takes the input without padding as DBA is able to process sequences in various lengths.

For the experiments on the VQA-v2 dataset, we use the ALBERT (Lan et al., 2020) to extract question features, resulting $\mathbb{R}^{768}$ embedding for every token in a sentence. We use the gird image features (Jiang et al., 2020) obtained from a ResNet-152 model (He et al., 2016) in the vision part. For the $i^{th}$ grid, it is represented as a feature as $x_i \in \mathbb{R}^{2048}$, with maximum 608 grid features. After cross-attention interactions, both language and vision parts perform intra-modality fusion following (Yu et al., 2019), and the final answer is predicted via addition.

For the performance on the state space model, we use the S4 (Gu et al., 2022) as backbone. Our goal is to improve efficiency of the state space model while maintaining its performance. Note that DBA first compresses the input sequence from $\mathbb{R}^{n \times d}$ to $\mathbb{R}^{d_p \times d}$, then processes compressed feature and finally restores the sequence to its original dimension $\mathbb{R}^{n \times d}$. Therefore, we could extract the compressed feature in DBA as the input of state space model to improve efficiency. We use one layer of DBA to compress sequence length of the input, and DBA is plugged after the first layer of the S4 model.

The detailed settings with hyper-parameters are listed in Tables 7, 8, 9, and 10.

Table 7: Experiment settings on the LRA.

| Task | Depth | Heads | $d$ | $d_{ffn}$ | $d_p$ | $d_{in}$ |
|------|-------|-------|-----|-----------|-------|----------|
| ListOps | 6 | 8 | 512 | 2048 | 16 | 24 |
| Text | 4 | 4 | 256 | 1024 | 16 | 24 |
| Retrieval | 4 | 4 | 128 | 512 | 16 | 24 |
| Image | 1 | 8 | 64 | 128 | 16 | 24 |
| Pathfinder | 1 | 4 | 128 | 128 | 16 | 24 |

Table 8: Experiment settings on the UEA multivariate time series classification archive.

| Task | Depth | Heads | $d$ | $d_{ffn}$ | $d_p$ | $d_{in}$ |
|------|-------|-------|-----|-----------|-------|----------|
| Ethanolconcentration | 1 | 8 | 64 | 256 | 16 | 24 |
| Facedetection | 3 | 8 | 128 | 256 | 16 | 24 |
| Handwriting | 1 | 8 | 128 | 256 | 16 | 24 |
| Heartbeat | 1 | 8 | 64 | 256 | 16 | 24 |
| Japanesevowels | 3 | 8 | 128 | 256 | 16 | 24 |
| Pems-Sf | 1 | 8 | 128 | 512 | 16 | 24 |
| Selfregulationscp1 | 3 | 8 | 128 | 256 | 16 | 24 |
| Selfregulationscp2 | 3 | 8 | 128 | 256 | 16 | 24 |
| Spokenarabicdigits | 3 | 8 | 128 | 256 | 16 | 24 |
| Uwavegesturelibrary | 3 | 16 | 256 | 256 | 16 | 24 |

Table 9: Experiment settings on the VQA-v2.

| Task | Depth | Heads | $d$ | $d_{ffn}$ | $d_p$ | $d_{in}$ |
|------|-------|-------|-----|-----------|-------|----------|
| VQA-v2 | 3 | 8 | 256 | 1024 | 24 | 64 |

Table 10: Experiment settings for the S4 with DBA optimization.

| Task | S4 Configuration | | DBA Configuration | | | | |
|---|---|---|---|---|---|---|---|
| | Depth | $d$ | Depth | Heads | $d$ | $d_p$ | $d_{in}$ |
| ListOps | 8 | 128 | 1 | 4 | 128 | 1024 | 64 |
| Text | 6 | 256 | 1 | 4 | 256 | 2048 | 128 |
| Retrieval | 6 | 256 | 1 | 4 | 256 | 2048 | 128 |
| Image | 6 | 512 | 1 | 4 | 512 | 512 | 256 |
| Pathfinder | 6 | 256 | 1 | 4 | 256 | 512 | 128 |

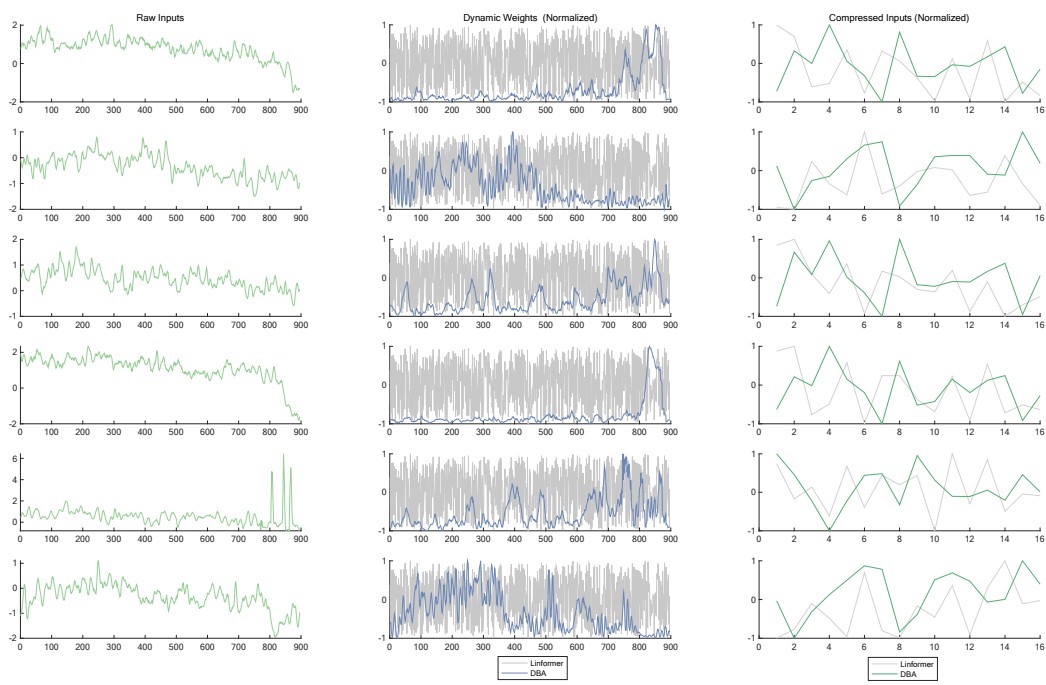

Figure 3: Visualization of sequence length compression matrices. The first column is the input sequence. The second column compares the dynamic sequence length projection matrix in DBA and the learned input invariant projection matrix in Linformer. The third column illustrates the compressed input by DBA and Linformer, respectively. Different rows represent different input samples.

## A.4 VISUALIZATION OF DYNAMIC SEQUENCE LENGTH PROJECTION MATRICES

We visualized the dynamic sequence lengths compression matrices in DBA and compared them with the input invariant compression matrices in Linformer on the Selfregulationscp1 task (Birbaumer et al., 1999), as shown in Figure 3. The Selfregulationscp1 record the EEG data and is one of the UEA multivariate time series classification archives. Results show that the sequence length projection matrix is determined by the input, which highlights values in different positions for different inputs, while Linformer concentrates on the same position for different samples. In addition, the sequence compression matrices in DBA are "smoother" between adjacent positions and have a more noticeable trend than the compression matrices in Linformer. From the characteristics of Selfregulationscp1 task and how people diagnose such diseases (Birbaumer et al., 1999), the concentrated position shall be different for different inputs and share more coherent trends between adjacent points like in DBA rather than oscillate. The DBA process the signals more human-like and could achieve higher performance, demonstrating the superiority of dynamic sequence length projection matrices.

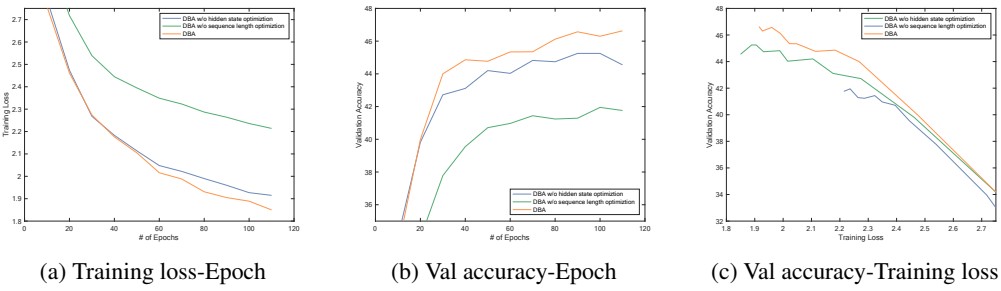

(a) Training loss-Epoch       (b) Val accuracy-Epoch       (c) Val accuracy-Training loss

Figure 4: Comparison of training loss and validation accuracy of DBA with ablation variants, including DBA without hidden state dimension compression and DBA without sequence length compression.

## A.5 ABLATION EXPERIMENTS

We conduct ablation studies on the LRA dataset to investigate the influence of $d_p$ and $d_{in}$ on efficiency and final performance, as shown in Tables 11 and 12. Results show that both sequence length and hidden state dimension compression contribute to the efficiency, where the sequence length compression contributes a higher speed-up ratio with the increase of sequence length (from 1.0x to 5.9x speed-up with 256-4k sequence length), and the compression in hidden state dimension contributes to dedicate speed-up rate (around 1.1x) for all inputs with different length. We also investigate the different settings of $d_p$ and $d_{in}$ to the efficiency, and DBA has faster speed with less memory consumption with the decrease of $d_p$ and $d_{in}$.

Table 12 illustrates $d_p$ and $d_{in}$ to the final performance. Results show that DBA could achieve a similar performance compared with the counterparts without sequence length or hidden state dimension compression on three out of five tasks on the LRA dataset, which is consistent with our theory that the optimization in DBA is either lossless or with high-order small amount error. The DBA achieves higher performance on the image and pathfinder tasks, which we believe the optimization in DBA contributes to the generation capability and ease of optimization. The DBA achieves higher validation accuracy under the same training loss and has faster coverage speed, as detailed shown in Figure 4. Note that DBA is faster and uses less memory than the ablation counterparts without sequence length or hidden state dimension compression, demonstrating the efficiency and effectiveness of DBA.

Different settings of $d_p$ and $d_{in}$ to the final performance are also illustrated in Table 12. We set $d_{in} = 24$ for the experiment of $d_p$ and $d_p = 16$ for the experiments of $d_{in}$. Increasing the $d_p$ from 8 to 128, DBA achieves higher performances, especially on the retrieval, image, and pathfinder tasks, while keeping similar performances on the other two tasks. However, the performance on the pathfinder task drops when increasing the $d_p$ to 256. Increasing the $d_{in}$ from 12 to 64, DBA also achieves higher performances on the same three tasks. For the balance between performance and efficiency, we set $d_p = 16$ and $d_{in} = 24$ for all tasks on the LRA dataset, as shown in Table 7.

Table 11: Inference speed and peak memory consumption of DBA with different $d_p$ and $d_{in}$ on byte-level text classification with various sequence lengths (1K, 2K, 3K, and 4K). The average performances are listed on the right. DBA coverage faster than the controlled groups with higher validation accuracy in the same training loss.

| | Model | Speed ↑ | | | | | | Peak Memory Usage ↓ | | | | | | Avg. ↑ |
| | | 256 | 512 | 1k | 2k | 3k | 4k | 256 | 512 | 1k | 2k | 3k | 4k | |
|---|---|---|---|---|---|---|---|---|---|---|---|---|---|---|
| | Vanilla Transformer | 1.0 | 1.0 | 1.0 | 1.0 | 1.0 | 1.0 | 1.0 | 1.0 | 1.0 | 1.0 | 1.0 | 1.0 | 58.57 |
| | w/o optimize $n$ | 1.1 | 1.1 | 1.1 | 1.1 | 1.1 | 1.1 | 0.96 | 0.96 | 0.96 | 0.96 | 0.98 | 0.99 | 59.75±0.24 |
| | 256 | 1.0 | 1.0 | 1.1 | 1.2 | 2.6 | 3.8 | 1.00 | 0.97 | 0.58 | 0.31 | 0.25 | 0.16 | 62.17±0.20 |
| $d_p$ | 64 | 1.0 | 1.3 | 1.3 | 1.8 | 3.9 | 5.6 | 0.87 | 0.72 | 0.42 | 0.21 | 0.17 | 0.10 | 62.26±0.10 |
| | 16 | 1.1 | 1.4 | 1.4 | 2.0 | 4.1 | 6.1 | 0.84 | 0.66 | 0.38 | 0.19 | 0.15 | 0.09 | 62.21±0.21 |
| | 8 | 1.2 | 1.4 | 1.5 | 2.1 | 4.2 | 6.4 | 0.83 | 0.65 | 0.37 | 0.19 | 0.15 | 0.09 | 61.73±0.28 |
| | w/o optimize $d$ | 1.0 | 1.1 | 1.4 | 1.9 | 4.0 | 5.9 | 0.85 | 0.68 | 0.39 | 0.20 | 0.16 | 0.09 | 60.95±0.32 |
| $d_{in}$ | 64 | 1.1 | 1.4 | 1.4 | 1.9 | 4.1 | 5.9 | 0.84 | 0.67 | 0.38 | 0.20 | 0.16 | 0.09 | 62.26±0.16 |
| | 24 | 1.1 | 1.4 | 1.4 | 2.0 | 4.1 | 6.1 | 0.84 | 0.66 | 0.38 | 0.19 | 0.15 | 0.09 | 62.21±0.21 |
| | 12 | 1.2 | 1.4 | 1.4 | 2.0 | 4.2 | 6.3 | 0.84 | 0.66 | 0.38 | 0.19 | 0.15 | 0.09 | 61.37±0.18 |

Table 12: Performance of DBA on the LRA benchmark with different $d_p$ and $d_{in}$.

| | Model | ListOps ↑ | Text ↑ | Retrieval ↑ | Image ↑ | Pathfinder ↑ | Avg. ↑ |
|---|---|---|---|---|---|---|---|
| | Vanilla Transformer | 36.37 | 64.27 | 78.38 | 42.44 | 71.40 | 58.57 |
| | w/o optimize $n$ | 38.08±0.23 | 66.00±0.10 | 78.53±0.04 | 41.76±0.54 | 74.37±0.28 | 59.75±0.24 |
| | 256 | 37.53±0.08 | 66.20±0.14 | 80.81±0.14 | 48.03±0.17 | 78.30±0.46 | 62.17±0.20 |
| $d_p$ | 64 | 37.40±0.05 | 66.17±0.12 | 80.69±0.09 | 46.98±0.09 | 80.05±0.17 | 62.26±0.10 |
| | 16 | 38.10±0.40 | 66.25±0.04 | 80.64±0.01 | 46.51±0.50 | 79.56±0.10 | 62.21±0.21 |
| | 8 | 37.88±0.33 | 66.06±0.13 | 80.37±0.19 | 45.70±0.30 | 78.65±0.46 | 61.73±0.28 |
| | w/o optimize $d$ | 37.55±0.25 | 65.92±0.10 | 79.98±0.19 | 45.39±0.34 | 75.90±0.73 | 60.95±0.32 |
| $d_{in}$ | 64 | 37.50±0.05 | 66.19±0.09 | 80.22±0.13 | 47.64±0.16 | 79.76±0.35 | 62.26±0.16 |
| | 24 | 38.10±0.40 | 66.25±0.04 | 80.64±0.01 | 46.51±0.50 | 79.56±0.10 | 62.21±0.21 |
| | 12 | 37.65±0.15 | 66.18±0.10 | 78.14±0.13 | 46.24±0.36 | 78.65±0.18 | 61.37±0.18 |

