# OpenReview forum: "DBA: Efficient Transformer with Dynamic Bilinear Low-Rank Attention"
_ICLR.cc/2023/Conference — Submitted to ICLR 2023_

### Official Review · Reviewer_YUik · 2022-10-19

**Confidence:** 5
**Correctness:** 1
**Technical Novelty And Significance:** 1
**Empirical Novelty And Significance:** 1
**Recommendation:** 3

**Clarity, Quality, Novelty And Reproducibility:**

As I described above, I found the paper is not easy to read given the writing mistakes (clarity), has mathematical flaws in the analysis (quality), leads to poor experimental results compared to SoTA (novelty).

**Strength And Weaknesses:**

I believe the paper is far below the paper acceptance threshold and here are several critical points that needs to be essentially addressed.

[Writing Quality]

Thr writing is rather poor and there are plenty of mistakes. I list some of them as below but there are much more in the paper.

1. In most places, term "the" is not correctly used.
2. In section 3.1, what is non-destructively?
3. In section 3.1, what is conditional extropy?
4. In section 3.2, why you call Johnson–Lindenstrauss lemma as "JS lemma"
5. In section 3.2, "an d \times d matrix"

[Math]

A critical flaw occurs in equation (6). It is ok to assume that QK is a low-rank matrix. However, P = softmax(QK), and softmax is a non-linear operator. There is no guarantee that a low-rank matrix passing through a non-linear operator is still a low-rank matrix. Furthermore, in practice, for the softmax function, P is usually a high-rank function. I recommend the authors have a try. Given this, we cannot get P=WrP'Wc, where all of them are low-rank.

Note that in any case, you can always use JK lemma and random projections. But such an analysis has already been used in Linformer.

[Experiment]

The author missed important effcient Transformer S4 ("Efficiently Modeling Long Sequences with Structured State Spaces"). In all of the LRA tasks, S4 is roughly better than the proposed DBA for 25 points. I have little reason to vote a paper for acceptance given the significant performance gap.



**Summary Of The Paper:**

This work suggests that the attention matrix, a key module in Transformer but computationally inefficient, can be approximated by low-rank matrix multiplications. Given the arguments, the authors proposed a new efficient Transformer variant using random projections. Experiments, including popularly used LRA benchmarks, are conducted to analyze the empirical performance of the model with comparisons to several baselines.

**Summary Of The Review:**

I believe proposing efficient Transformer is essential but the current quality of the work (writing, analysis, empirical results) are not ready to publish in the venue, and I recommend rejection.

---

> ### Author Response · Authors · 2022-11-18
> **Response to Reviewer YUik (Part 1)**
>
> **Q1: In most places, term "the" is not correctly used.**
>
> **A1:** Thanks for your comment. We polished the writing and rechecked the grammar.
>
> **Q2: In section 3.1, what is non-destructively?**
>
> **A2:** Thanks for your comment and sorry for the confusion. “Non-destructively” indicates the sequence length compression techniques in DBA do not theoretically cause any loss of information. We have replaced “non-destructively” with “losslessly” to avoid confusion.
>
> **Q3: In section 3.1, what is conditional extropy?**
>
> **A3:** Thanks for your comment and sorry for the misspelling. We have corrected the spelling to “conditional entropy”. In information theory, conditional entropy quantifies the amount of information needed to describe the outcome of a random variable $Y$ given that the value of another random variable $X$ is known.
>
> **Q4: In section 3.2, why you call Johnson–Lindenstrauss lemma as "JS lemma"**
>
> **A4:** Thanks for your comment and sorry for the confusion. We cancel the use of abbreviation for Johnson–Lindenstrauss lemma to avoid misunderstanding.
>
> **Q5: In section 3.2, "an d \times d matrix"**
>
> **A5:** Thanks for your comment. We replaced the sentence "Let $R$ be an $d\times d_{in}$ matrix" with "Let $R \in\mathbb{R}^{d\times d_{in}}$".
>
> **Q6: A critical flaw occurs in equation (6). It is ok to assume that QK is a low-rank matrix. However, P = softmax(QK), and softmax is a non-linear operator. There is no guarantee that a low-rank matrix passing through a non-linear operator is still a low-rank matrix. Furthermore, in practice, for the softmax function, P is usually a high-rank function. I recommend the authors have a try. Given this, we cannot get P=WrP'Wc, where all of them are low-rank.**
>
> **A6:** Thanks for your comment and sorry for the confusion. It is true that the non-linear operator $\operatorname{softmax}$ could change the rank of the input matrix. However, $\operatorname{softmax}$ operator does not bring any information gain compared to the original matrix. Instead, $\operatorname{softmax}$ may cause the loss of information. Here, the expression can be written as:
>
> $\operatorname{H}(\operatorname{softmax}(QK^T); QK^T)=0$
>
> $\operatorname{H}(QK^T; \operatorname{softmax}(QK^T))>0$
>
> where $\operatorname{H}()$ indicates conditional entropy.
>
> An easy way to understand the aforementioned theory is assuming we have two matrices $A$ and $B$ satisfy $B=\operatorname{softmax}(A)$. We could always obtain deterministic and unique matrix $B$ if we know $A$. However, we could not reconstruct the deterministic and unique $A$ if we just know $B$.
>
> Hence, **the $QK^T$ contains all the information regarding the $\operatorname{softmax}(QK^T)$**. Notice that we have:
>
> $\operatorname{H}(\operatorname{softmax}(QK^T); QK^T)=0$
>
> $H(QK^T; \operatorname{basis_r}(\operatorname{basis_c}(QK^T))), W_r’, W_c’)=0$
>
> Therefore, the { $\operatorname{basis_r}(\operatorname{basis_c}(QK^T)))$, $W_r’$, $W_c’$ } contains all the information that $P=\operatorname{softmax}(QK^T/\sqrt{d})$ has. In practice, we use $P_\phi=W_r^\prime{P_\phi}^\prime{W_c^\prime}^T=W_r^\prime\left(\operatorname{softmax}\left(\frac{Q_l K_l^T}{\sqrt{d}}\right)\right){W_c^\prime}^T $ to reconstruct $P$, where $Q_l\in\mathbb{R}^{d_p\times d}$, $K_l\in\mathbb{R}^{d_p\times d}$, $W_r^\prime\in\mathbb{R}^{n\times d_p}$, and $W_c^\prime\in\mathbb{R}^{n\times d_p}$. The reconstruction process, including variables $Q_l$, $K_l$, $W_r’$, and $W_c’$ are learned through the training process. We have modified Section 3.1 for better articulation between the theory and implementation details.

---

> > ### Author Response · Authors · 2022-11-18
> > **Response to Reviewer YUik (Part 2)**
> >
> > **Q7: Note that in any case, you can always use JK lemma and random projections. But such an analysis has already been used in Linformer.**
> >
> > **A7:** Thanks for your comment. We highlight the key differences between DBA and Linformer, as shown in the following Table. The DBA compress sequence length with dynamic projections determined by input for all $Q$, $K$, $V$ in Transformer and focus on the impact of hidden state dimension, while the Linformer compresses sequence length with input invariant patterns and ignores the impact of hidden state dimension.
> >
> > Comparsion of DBA with Linformer.
> >
> > |Model|Sequence Length Compression Matrix|Process Sequence in Various Length|Compressed Variables in Transformer|Hidden State Dimension Compression|Applied Models|Speed $\uparrow$|Average Performance $\uparrow$|
> > |:-:|:-:|:-:|:-:|:-:|:-:|:-:|:-:|
> > |Linformer|Learned parameters (input invarient)|-|$K, V$|-|Transformer|5.5x|56.47|
> > |**DBA**|Dynamically determined by the input|$ \checkmark$|$Q$, $K$, $V$|$ \checkmark$|Transformer, State Space Model|6.1x|**62.21±0.21**|
> >
> > Due to the aforementioned differences, DBA has three main benefits over the previous low-rank based approaches. Firstly, DBA could concentrate on different positions for different inputs to best preserve the most informative parts of a sequence, as analyzed in Section 1 and verified from the performance in Sections 4.2-4.4 and the visualization results in Section A.4. The DBA outperforms the Linformer over 5% in terms of average accuracy (62.29 verses 56.47) on LRA dataset and achieves higher efficiency in both speed and memory consumption perspectives. Secondly, DBA could process sequences in various lengths as the dimensions of sequence length compression matrices are also determined by the input. Thirdly, DBA could achieve state-of-the-art performance with high efficiency over various sequence length conditions due to jointly considering the sequence length and hidden state dimension, as detailed illustrated in Sections 4.1 and 4.3.
> >
> > The DBA extends the Johnson–Lindenstrauss lemma to show that the multiplication between $Q$ and $K^T$ can be approximated with high-order small amount error, which mitigates the impact of hidden state dimension $d$ on efficiency. The Linformer applied the Johnson–Lindenstrauss lemma between $\operatorname{softmax}(QK^T/\sqrt{d})$ and $V$ to compress the sequence length dimension using input invariant parameters. Due to the abovementioned differences, the analyses and resulting models have differences, leading to three major advantages over Linformer. To address the concern, we move the proof parts of Section 3.2 to Section A.1 and only keep the main conclusion in Section 3.2.
> >
> > **Q8: The author missed important effcient Transformer S4 ("Efficiently Modeling Long Sequences with Structured State Spaces"). In all of the LRA tasks, S4 is roughly better than the proposed DBA for 25 points. I have little reason to vote a paper for acceptance given the significant performance gap.**
> >
> > **A8:** Thanks for your contributing comment. As a different approach from Transformer, the state space model is a foundational scientific model used in control theory and computational neuroscience. Recently, it was introduced as a sequence model and has achieved promising results, especially for long sequence modeling. The state space model takes the similar input $X \in\mathbb{R}^{n\times d}$ as the Transformer when processing the sequence, with its speed and memory consumption much influenced by the sequence length $n$.
> >
> > Note that DBA first compresses the input sequence from $\mathbb{R}^{n\times d}$ to $\mathbb{R}^{d_p\times d}$, then processes the compressed feature and finally restores the sequence to its original dimension $\mathbb{R}^{n\times d}$. Here, we could extract the compressed feature in DBA as the input of the state space model to improve efficiency. We extend DBA to one of the state space models S4 [1] by plugging DBA into the S4 model, as detailed described in Sections 4.5 and A.3. Results show that the S4 with DBA optimization could achieve **1.4x average speed boost and 0.8x average memory consumption with competitive performance compared to the original S4 model**, as shown in Table 5, demonstrating the universality of DBA.
> >
> > Efficiency and performance of DBA with S4 on LRA dataset.
> >
> > |||S4|||S4 + DBA||
> > |:-:|:-:|:-:|:-:|:-:|:-:|:-:|
> > ||Accuracy $\uparrow$|Speed $\uparrow$|Memory $\downarrow$|Accuracy $\uparrow$|Speed $\uparrow$|Memory $\downarrow$|
> > |ListOps|59.60|1.0|1.0|59.70|1.3|0.8|
> > |Text|86.20|1.0|1.0|85.40|1.2|0.9|
> > |Retrieval|90.90|1.0|1.0|91.39|1.4|0.9|
> > |Image|87.28|1.0|1.0|86.90|1.5|0.8|
> > |Pathfinder|94.20|1.0|1.0|93.98|1.4|0.8|
> > |Avg.|83.64|1.0|1.0|83.47|1.4|0.8|
> >
> > [1] Albert Gu, Karan Goel, and Christopher Re. Efficiently modeling long sequences with structured
> > state spaces. In $\textit{International Conference on Learning Representations}$, 2022.

---

> > > ### Author Response · Authors · 2022-11-18
> > > **Response to Reviewer YUik (Part 3)**
> > >
> > > **Q9: I believe proposing efficient Transformer is essential but the current quality of the work (writing, analysis, empirical results) are not ready to publish in the venue, and I recommend rejection.**
> > >
> > > **A9:** Thanks for your comment. We address the concerns about writing, analysis, and empirical results point by point and have modified the manuscript accordingly.

---

> ### Author Response · Authors · 2022-11-27
> **Response to Reviewer YUik**
>
> Dear reviewer, we have tried to address your constructive comments in the earlier response. Please feel free to contact us if there are any further questions or suggestions. Discussions are always welcome.

---

> ### Author Response · Authors · 2022-11-30
> **Response to Reviewer YUik**
>
> Dear reviewer,
>
> Thank you for your valuable comments and suggestions in reviewing our manuscript. The constructive comments you addressed have enabled us to disseminate our manuscript to the higher possible quality, and we have made extensive revisions according. Please let us know if there are any further comments or suggestions. Just as a reminder, the discussion stage is due by Dec 12, 2022.
>
> We are really grateful for your thorough review and looking forward to hearing from you in due course.

---

> ### Author Response · Authors · 2022-12-11
> **Reminder for Reviewer YUik**
>
> Dear reviewer,
>
> Thank you for your constructive comments and suggestions in reviewing our manuscript. We have made extensive revisions according to your valuable comments point by point. Please let us know if there are any further suggestions or comments. Just as a reminder, the discussion stage is due by Dec 12, 2022.
>
> We are really grateful for your thorough review and are looking forward to hearing from you as soon as possible.

---

### Official Review · Reviewer_PZVd · 2022-10-24

**Confidence:** 5
**Clarity, Quality, Novelty And Reproducibility:** The presentation needs to be improved…
**Correctness:** 3
**Technical Novelty And Significance:** 3
**Empirical Novelty And Significance:** 3
**Recommendation:** 6

**Strength And Weaknesses:**

Pros:
- The learnable projection in the paper is novel and squeezes the computation not only in terms of sequence length but also in terms of hidden dimension. It is encouraging to see that simply choosing d_p and d_in to be fixed suffices for achieving good performance.
- The empirical results are promising compared to other linear attention-based models.
Cons:
- The presentation in Section 3.1 is a little confusing because the W_r' and W_c' depend on the input QK, while Equation (4) makes readers feel they are given.
- Table 2 in the paper is a little misleading because the methods are not sorted by any meaningful order (e.g., Avg)
- (Minor) the performance cannot be comparable with new models based on state-space models. It might be helpful to apply similar ideas as this paper to those models to achieve further gain.

**Summary Of The Paper:**

This paper proposed to learn the projection matrices for reducing the dimension of intermediate matrices in Transformer models and achieves significant speedup while achieving better performance than previous efficient attentions.

**Summary Of The Review:**

In summary, the method in this paper is novel and interesting, while some minor modifications are needed to clarify the paper.

---

> ### Author Response · Authors · 2022-11-18
> **Response to Reviewer PZVd**
>
> **Q1: The presentation in Section 3.1 is a little confusing because the $W_r'$ and $W_c'$ depend on the input $QK$, while Equation (4) makes readers feel they are given.**
>
> **A1:** Thanks for your comment and sorry for the confusion. We revised Section 3.1 by adding " $W_r^\prime$ and $W_c^\prime$ are the reconstruction coefficients for row and column, which values and dimensions are determined by $QK^T$".
>
> **Q2: Table 2 in the paper is a little misleading because the methods are not sorted by any meaningful order (e.g., Avg).**
>
> **A2:** Thanks for your comment. We sorted Table 2 by the average accuracy.
>
> **Q3: (Minor) the performance cannot be comparable with new models based on state-space models. It might be helpful to apply similar ideas as this paper to those models to achieve further gain.**
>
> **A3:**	Thanks for your contributing comment. As a different approach from Transformer, the state space model is a foundational scientific model used in control theory and computational neuroscience. Recently, it was introduced as a sequence model and has achieved promising results, especially for long sequence modeling. The state space model takes the similar input $X \in\mathbb{R}^{n\times d}$ as the Transformer when processing the sequence, with its speed and memory consumption much influenced by the sequence length $n$.
>
> Note that DBA first compresses the input sequence from $\mathbb{R}^{n\times d}$ to $\mathbb{R}^{d_p\times d}$, then processes the compressed feature and finally restores the sequence to its original dimension $\mathbb{R}^{n\times d}$. Here, we could extract the compressed feature in DBA as the input of the state space model to improve efficiency. We extend DBA to one of the state space models S4 [1] by plugging DBA into the S4 model, as detailed described in Sections 4.5 and A.3. Results show that the S4 with DBA optimization could achieve **1.4x average speed boost and 0.8x average memory consumption with competitive performance compared to the original S4 model**, as shown in Table 5, demonstrating the universality of DBA.
>
> Efficiency and performance of DBA with S4 on LRA dataset.
>
> |||S4|||S4 + DBA||
> |:-:|:-:|:-:|:-:|:-:|:-:|:-:|
> ||Accuracy $\uparrow$|Speed $\uparrow$|Memory $\downarrow$|Accuracy $\uparrow$|Speed $\uparrow$|Memory $\downarrow$|
> |ListOps|59.60|1.0|1.0|59.70|1.3|0.8|
> |Text|86.20|1.0|1.0|85.40|1.2|0.9|
> |Retrieval|90.90|1.0|1.0|91.39|1.4|0.9|
> |Image|87.28|1.0|1.0|86.90|1.5|0.8|
> |Pathfinder|94.20|1.0|1.0|93.98|1.4|0.8|
> |Avg.|83.64|1.0|1.0|83.47|1.4|0.8|
>
> [1] Albert Gu, Karan Goel, and Christopher Re. Efficiently modeling long sequences with structured
> state spaces. In $\textit{International Conference on Learning Representations}$, 2022.
>
> **Q4: The presentation needs to be improved, and please refer to the cons.**
>
> **A4:**	Thanks for your comments. We polished the writing and rechecked the grammar. The cons are addressed point by point.
>
> **Q5: In summary, the method in this paper is novel and interesting, while some minor modifications are needed to clarify the paper.**
>
> **A5:** Thanks for your positive comment. We have made modifications according to your constructive comments.

---

### Official Review · Reviewer_9GWA · 2022-10-25

**Confidence:** 4
**Correctness:** 3
**Technical Novelty And Significance:** 3
**Empirical Novelty And Significance:** 3
**Recommendation:** 6

**Clarity, Quality, Novelty And Reproducibility:**

Quality:
- The results in Tables 3 and 4 are very close to the ones obtained by related approaches. It is hard to assess the quality of the results without confidence intervals.

Clarity:
- It is unknown how DBA was applied for dynamic length inputs in section 4.3.
- The writing is casual in some parts. E.g., "A lot of studies ..." or "... by digging into the ..."
- Overall, I think the writing can be improved.

**Strength And Weaknesses:**

Things that I liked in this paper:

- The authors explain all the matrix transformation steps in detail, making the linear algebra math easy to follow in section 3.1.
- The idea of using information about the input sequence before projecting it to a shorter sequence.
- The results are impressive.

However, there are many things that I disliked:

- The overall method is not novel, and most central ideas were proposed in previous approaches (Linformer and Performer).
- The introduction is very repetitive.
- Section 3.2 seems unnecessary and confusing.
- The experimental section is uninformative, especially for sections 4.3 and 4.4.
- Some math terms are not defined. For example, what is "Linear" in Equation 18? Is it needed? What is the actual function used as $\psi$?



**Summary Of The Paper:**

This paper presents a new low-rank method, called DBA, to approximate the attention mechanism used in transformer layers. It has two main differences from previous approaches:

1. DBA maps the input sequences $Q, K \in \mathbb{R}^{n \times d}$ to shorter sequences $Q', K' \in \mathbb{R}^{d_p \times d}$ via a linear transformation that captures information from the input matrix itself, e.g., $Q' = \phi(WQ^\top)Q$, where $W \in \mathbb{R}^{d_p \times d}$ are learnable parameters.
2. DBA also projects queries and key vectors to lower dimensionalities ($d_{in} \ll d$)

Results show that DBA is faster and consumes less memory than related approaches while achieving strong predictive accuracies on several tasks.



**Summary Of The Review:**

This paper proposes a new efficient alternative to attention in transformers. The proposed method achieves impressive results on several tasks while being time and memory efficient. However, I believe this paper can be improved in terms of clarity and quality, especially regarding the experimental section.

---

> ### Author Response · Authors · 2022-11-18
> **Response to Reviewer 9GWA (Part 1)**
>
> **Q1: The overall method is not novel, and most central ideas were proposed in previous approaches (Linformer and Performer).**
>
> **A1:** Thanks for your comments. We highlight the key differences between DBA and Linformer in the following Table and describe the differences compared with Performer.
>
> Comparsion of DBA with Linformer.
>
> |Model|Sequence Length Compression Matrix|Process Sequence in Various Length|Compressed Variables in Transformer|Hidden State Dimension Compression|Applied Models|Speed $\uparrow$|Average Performance $\uparrow$|
> |:-:|:-:|:-:|:-:|:-:|:-:|:-:|:-:|
> |Linformer|Learned parameters (input invarient)|-|$K, V$|-|Transformer|5.5x|56.47|
> |**DBA**|Dynamically determined by the input|$ \checkmark$|$Q$, $K$, $V$|$ \checkmark$|Transformer, State Space Model|6.1x|**62.21±0.21**|
>
> The differences between DBA and Linformer are both in theory and model architecture perspectives. From the theory perspective, we first theoretically show that the sequence length can be compressed losslessly from a novel perspective of the information theory. In addition, we demonstrate that the hidden state dimension can be approximated by extending the Johnson–Lindenstrauss lemma with high-order small amount error.
>
> From the model architecture perspective, the sequence length compression matrices in DBA are dynamically determined by the input rather than in pre-determined or input invariant patterns like Linformer. Furthermore, DBA further compresses hidden state dimension to achieve further efficiency.
>
> Due to the aforementioned differences, DBA has three main benefits over previous low-rank based approaches. Firstly, DBA could concentrate on different positions for different inputs to best preserve the most informative parts of a sequence, as analyzed in Section 1 and verified from the performance in Sections 4.2-4.4 and the visualization results in Section A.4. The DBA outperforms the Linformer over 5% in terms of average accuracy (62.29 verses 56.47) on LRA dataset and achieves higher efficiency in both speed and memory consumption perspectives. Secondly, DBA could process sequences in various lengths as the dimensions of sequence length compression matrices are also determined by the input. Thirdly, DBA could achieve state-of-the-art performance with high efficiency over various sequence length conditions due to jointly considering the sequence length and hidden state dimension, as detailed illustrated in Sections 4.1 and 4.3.
>
> As for the comparison with Performer, although both DBA and Performer achieve linear efficiency through matrix optimization, there are many differences in the model architecture and theory. From the model architecture perspective, DBA optimizes the sequence length with dynamic projection matrics for all $K$, $Q$, $V$ in Transformer, while the Performer did not compress the sequence length dimension. Instead, they achieved linear efficiency with optimized $\operatorname{softmax}$ kernel and computation order. From the theoretical perspective, DBA optimizes the Transformer from a novel perspective of information theory, while Performer analyzes from the perspective of random features.
>
> **Q2: The introduction is very repetitive.**
>
> **A2:** Thanks for your comment. We streamlined the introduction, especially the background in Section 1.
>
> **Q3: Section 3.2 seems unnecessary and confusing.**
>
> **A3:** Thanks for your comment and sorry for the confusion. In Section 3.2, we extend the Johnson–Lindenstrauss lemma to prove that the hidden state dimension can be compressed with high-order small amount error in the results. By jointly optimizing the sequence length and hidden state dimension, DBA could achieve state-of-the-art performance with high efficiency over various sequence lengths. We keep the main conclusion in Section 3.2 while moving the proof parts to Section A.1 for more clarity.

---

> > ### Author Response · Authors · 2022-11-18
> > **Response to Reviewer 9GWA (Part 2)**
> >
> > **Q4: The experimental section is uninformative, especially for sections 4.3 and 4.4.**
> >
> > **A4:** Thanks for your contributing comment.
> >
> > We added ablation experiments regarding the efficiency and performance of DBA on the LRA dataset, as shown in Section A.5 and Tables 11-12. Results show that on three of the five tasks, DBA can achieve similar performance compared to its ablation counterparts without sequence length or hidden state dimension compression, which is consistent with the proposed theory that sequence length compression is lossless and the hidden state dimension is approximated with high-order small amount error. The DBA achieves higher performance on the Image and Pathfinder datasets. We believe the principle of DBA contributes to the generalizability and ease of optimization, which achieves higher validation accuracy under the same training loss and faster coverage speed than the ablation counterparts, as shown in Figure 4. Note that DBA is faster and uses less memory than the ablation counterparts, demonstrating the efficiency and effectiveness of DBA.
> >
> > Different settings regarding the compression rates for sequence length and hidden state dimension are also illustrated in Tables 11-12. Results show that DBA could achieve balance between efficiency and performance with proper configurations of $d_p$ and $d_{in}$.
> >
> > Inference speed and average performance of DBA with different $d_p$.
> >
> > ||256 $\uparrow$|512 $\uparrow$|1k $\uparrow$|2k $\uparrow$|3k $\uparrow$|4k $\uparrow$|Avg. $\uparrow$|
> > |:-:|:-:|:-:|:-:|:-:|:-:|:-:|:-:|
> > |Vanilla Transformer|1.0|1.0|1.0|1.0|1.0|1.0|58.57|
> > |w/o optimize $n$|1.1|1.1|1.1|1.1|1.1|1.1|59.75±0.24|
> > |256|1.0|1.0|1.1|1.2|2.6|3.8|62.17±0.20|
> > |64|1.0|1.3|1.3|1.8|3.9|5.6|62.26±0.10|
> > |16|1.1|1.4|1.4|2.0|4.1|6.1|62.21±0.21|
> > |8|1.2|1.4|1.5|2.1|4.2|6.4|61.73±0.28|
> >
> > Peak memory consumption and average performance of DBA with different $d_p$.
> >
> > ||256 $\downarrow$|512 $\downarrow$|1k $\downarrow$|2k $\downarrow$|3k $\downarrow$|4k $\downarrow$|Avg. $\uparrow$|
> > |:-:|:-:|:-:|:-:|:-:|:-:|:-:|:-:|
> > |Vanilla Transformer|1.0|1.0|1.0|1.0|1.0|1.0|58.57|
> > |w/o optimize $n$|0.96|0.96|0.96|0.96|0.98|0.99|59.75±0.24|
> > |256|1.00|0.97|0.58|0.31|0.25|0.16|62.17±0.20|
> > |64|0.87|0.72|0.42|0.21|0.17|0.10|62.26±0.10|
> > |16|0.84|0.66|0.38|0.19|0.15|0.09|62.21±0.21|
> > |8|0.83|0.65|0.37|0.19|0.15|0.09|61.73±0.28|
> >
> > Inference speed and average performance of DBA with different $d_{in}$.
> >
> > ||256 $\uparrow$|512 $\uparrow$|1k $\uparrow$|2k $\uparrow$|3k $\uparrow$|4k $\uparrow$|Avg. $\uparrow$|
> > |:-:|:-:|:-:|:-:|:-:|:-:|:-:|:-:|
> > |Vanilla Transformer|1.0|1.0|1.0|1.0|1.0|1.0|58.57|
> > |w/o optimize $d$|1.0|1.1|1.4|1.9|4.0|5.9|60.95±0.32|
> > |64|1.1|1.4|1.4|1.9|4.1|5.9|62.26±0.16|
> > |24|1.1|1.4|1.4|2.0|4.1|6.1|62.21±0.21|
> > |12|1.2|1.4|1.4|2.0|4.2|6.3|61.37±0.18|
> >
> > Peak memory consumption and average performance of DBA with different $d_{in}$.
> >
> > ||256 $\downarrow$|512 $\downarrow$|1k $\downarrow$|2k $\downarrow$|3k $\downarrow$|4k $\downarrow$|Avg. $\uparrow$|
> > |:-:|:-:|:-:|:-:|:-:|:-:|:-:|:-:|
> > |Vanilla Transformer|1.0|1.0|1.0|1.0|1.0|1.0|58.57|
> > |w/o optimize $d$|0.85|0.68|0.39|0.20|0.16|0.09|60.95±0.32|
> > |64|0.84|0.67|0.38|0.20|0.16|0.09|62.26±0.16|
> > |24|0.84|0.66|0.38|0.19|0.15|0.09|62.21±0.21|
> > |12|0.84|0.66|0.38|0.19|0.15|0.09|61.37±0.18|
> >
> > Performance of DBA with different $d_p$ on the LRA benchmark.
> >
> > Models|ListOps $\uparrow$|Text $\uparrow$|Retrieval $\uparrow$|Image $\uparrow$|Pathfinder $\uparrow$|Avg. $\uparrow$|
> > |:-:|:-:|:-:|:-:|:-:|:-:|:-:|
> > w/o optimize $n$|38.08±0.23|66.00±0.10|78.53±0.04|41.76±0.54|74.37±0.28|59.75±0.24|
> > 256|37.53±0.08|66.20±0.14|80.81±0.14|48.03±0.17|78.30±0.46|62.17±0.20|
> > 64|37.40±0.05|66.17±0.12|80.69±0.09|46.98±0.09|80.05±0.17|62.26±0.10|
> > 16|38.10±0.40|66.25±0.04|80.64±0.01|46.51±0.50|79.56±0.10|62.21±0.21|
> > 8|37.88±0.33|66.06±0.13|80.37±0.19|45.70±0.30|78.65±0.46|61.73±0.28|
> >
> > Performance of DBA with different $d_{in}$ on the LRA benchmark.
> >
> > Models|ListOps $\uparrow$|Text $\uparrow$|Retrieval $\uparrow$|Image $\uparrow$|Pathfinder $\uparrow$|Avg. $\uparrow$|
> > |:-:|:-:|:-:|:-:|:-:|:-:|:-:|
> > w/o optimize $d$|37.55±0.25|65.92±0.10|79.98±0.19|45.39±0.34|75.90±0.73|60.95±0.32|
> > 64|37.50±0.05|66.19±0.09|80.22±0.13|47.64±0.16|79.76±0.35|62.26±0.16|
> > 24|38.10±0.40|66.25±0.04|80.64±0.01|46.51±0.50|79.56±0.10|62.21±0.21|
> > 12|37.65±0.15|66.18±0.10|78.14±0.13|46.24±0.36|78.65±0.18|61.37±0.18|
> >
> > In addition, we visualize and analyze the dynamic sequence lengths compression matrices in DBA and compare them with the input invariant compression matrix in Linformer, as shown in Section A.4 and Figure 3. Results show that the DBA sequence length projection matrix is determined by the input, which highlights values in different positions for the different inputs, while the sequence length projection matrix in Linformer concentrates on the same position for different samples.
> >
> > Furthermore, we add more implementation details for Sections 4.3 and 4.4 in Section A.2.

---

> > > ### Author Response · Authors · 2022-11-18
> > > **Response to Reviewer 9GWA (Part 3)**
> > >
> > > **Q5: Some math terms are not defined. For example, what is "Linear" in Equation 18? Is it needed? What is the actual function used as $\varphi$.**
> > >
> > > **A5:**	Thanks for your comments and sorry for the confusion. $\operatorname{Linear()}$ is a fully connected layer and is defined as torch.nn.Linear() in Pytorch. We use the fully connected layer to bring more flexibility to the hidden state dimension. We set $\varphi$ as $\operatorname{softmax}$ function. The related sentences have been modified to avoid confusion.
> > >
> > > **Q6: The results in Tables 3 and 4 are very close to the ones obtained by related approaches. It is hard to assess the quality of the results without confidence intervals.**
> > >
> > > **A6:**	Thanks for your comment. We trained DBA on each task with 5 random seeds and reported the accuracy variance, as shown in Tables 2-4 and 12.
> > >
> > > **Q7: It is unknown how DBA was applied for dynamic length inputs in section 4.3.**
> > >
> > > **A7:**	Thanks for your comment and sorry for the confusion. The DBA is capable of processing sequences with various lengths because the sequence length compression matrices are dynamically determined by the input, including dimensions. Therefore, DBA is able to process sequences without fixed padding during implementation. During the training process, the batched inputs are padded to the max length similar to the Vanilla Transformer. We revised Section 3.3 and added more implementation details in Section A.2 to avoid confusion.
> > >
> > > **Q8: The writing is casual in some parts. E.g., "A lot of studies ..." or "... by digging into the ..."**
> > >
> > > **A8:**	Thanks for your comment. We polished the writing to avoid casualness.
> > >
> > > **Q9: Overall, I think the writing can be improved.**
> > >
> > > **A9:**	Thanks for your comment. We polished the writing and rechecked the grammar.
> > >
> > > **Q10: This paper proposes a new efficient alternative to attention in transformers. The proposed method achieves impressive results on several tasks while being time and memory efficient. However, I believe this paper can be improved in terms of clarity and quality, especially regarding the experimental section.**
> > >
> > > **A10:**	Thanks for your positive comments on the performance and efficiency of DBA. We address the clarity and quality of the experimental section in three aspects.
> > >
> > > Firstly, we added ablation experiments regarding the efficiency and performance of DBA on the LRA dataset, as shown in Section A.5 and Tables 11-12. Results show that DBA could achieve a similar performance compared to the ablation counterparts without sequence length or hidden state dimension compression on three out of five tasks, which is consistent with the proposed theory that the sequence length compression is lossless and the hidden state dimension is approximated with high-order small amount error. The DBA achieves higher performance on the Image and Pathfinder datasets. We believe the principle of DBA contributes to the generalizability and ease of optimization, which achieves higher validation accuracy under the same training loss and faster coverage speed than the ablation counterparts, as shown in Figure 4. In addition, different settings regarding the compression rates for sequence length and hidden state dimension are also illustrated in Tables 11-12. Results show that DBA could achieve balance between efficiency and performance with proper configurations.
> > >
> > > Secondly, we visualized the dynamic sequence lengths compression matrices in DBA and compared them with the input invariant compression matrix in Linformer, as shown in Figure 3. Results show that the DBA sequence length projection matrix is determined by the input, which highlights values in different positions for the different inputs, while the sequence length projection matrix in Linformer concentrates on the same position for different samples.
> > >
> > > Thirdly, we add more implementation details in Section A.2.

---

> > > > ### Comment · Reviewer_9GWA · 2022-11-21
> > > > **Response**
> > > >
> > > > I want to thank the authors for their careful and thorough responses.
> > > >
> > > > I still believe DBA has some strong connections with related approaches. However, I do recognize now its novel contributions. In particular, the ability to deal with variable-length sequences and the universality of DBA alongside its theoretical motivation is, in fact, a novel contribution of this work.
> > > >
> > > > Given the great work on analyzing the impact of $d_p$ and $d_{in}$ in the ablation studies, the interpretation of the dynamic sequence length matrices in Section A.4, and the overall clarification of my concerns, I am happy to increase my score.
> > > >
> > > > **It is unknown how DBA was applied for dynamic length inputs in section 4.3**
> > > >
> > > > Thanks for clarifying this. What is the impact of $\varphi$ in DBA? Have you tried transformations other than softmax?
> > > >
> > > > Small notes:
> > > > - $\boldsymbol{K}_{DBA}$ should be transposed in Equation 10.
> > > > - In Section 3.3, "input sequences propagate **thought** fully connected layers" -> "**through**"

---

> > > > > ### Author Response · Authors · 2022-11-27
> > > > > **Response to Reviewer 9GWA**
> > > > >
> > > > > **Many thanks for your positive comments about the novelty and contribution of DBA.**
> > > > >
> > > > > **Q1: Thanks for clarifying this. What is the impact of $\varphi$ in DBA? Have you tried transformations other than softmax?**
> > > > >
> > > > > **A1:** Thanks for your comment. We conduct an ablation experiment on the LRA dataset to illustrate the impact of $\varphi$, as shown in the following Table. Note that the $\varphi$ process the features $ZQ^T \in \mathbb{R}^{d_p\times n}$ and $ ZK^T \in \mathbb{R}^{d_p\times n}$, which dimensions change with the inputs. Therefore, the $\varphi$ shall be a function that is able to process features in different lengths. Results show that DBA could achieve state-of-the-art performance when setting $\varphi$ as softmax function. However, the DBA will either fail the training process due to NaN error or obtain very low accuracy without proper $\varphi$. We find that the model with sigmoid function achieves similar performances compared to the model with softmax function on two out of five tasks, including Text and Pathfinder tasks. The DBA with softmax function outperforms the ablation counterparts for the other three tasks. Therefore, we select softmax as $\varphi$ in DBA.
> > > > >
> > > > >
> > > > > Table. Ablation experiments of DBA with different $\varphi$. “NaN” denotes the model suffers NaN error during training process, and “NC” denotes the model does not converge.
> > > > >
> > > > >
> > > > > ||ListOps $\uparrow$|Text $\uparrow$|Retrieval $\uparrow$|Image $\uparrow$|Pathfinder $\uparrow$|Avg. $\uparrow$|
> > > > > |:-:|:-:|:-:|:-:|:-:|:-:|:-:|
> > > > > |w/o normalization|NaN|NaN|NaN|NaN|NaN|\-|
> > > > > |Layer Norm|NaN|NaN|NaN|43.88±0.58|52.01±0.51|\-|
> > > > > |Sigmoid|NC|66.16±0.06|79.78±0.37|44.14±0.08|79.80±0.12|\-|
> > > > > |Softmax|38.10±0.40|66.25±0.04|80.64±0.01|46.51±0.50|79.56±0.10|62.21±0.21|
> > > > >
> > > > > **Q2: 1. $K_{DBA}$ should be transposed in Equation 10. 2. In Section 3.3, "input sequences propagate thought fully connected layers" -> "through"**
> > > > >
> > > > > Thanks for your comment. We will revise the writing mistakes in the final version of our manuscript.
> > > > >
> > > > > Please feel free to contact us if there are any questions or suggestions. Discussions are always welcome.

---

### Official Review · Reviewer_tdpB · 2022-10-27

**Confidence:** 3
**Correctness:** 3
**Technical Novelty And Significance:** 2
**Empirical Novelty And Significance:** 2
**Recommendation:** 5

**Clarity, Quality, Novelty And Reproducibility:**

Overall the technical details are sufficient and the experiment results look promising. It would be better if the authors can further justify the novelty of the proposed method.

**Strength And Weaknesses:**

Strength:
* The technical details are presented clearly and with details
* The experiments include state-of-the-art methods as baselines
* The paper is structured well and easy to follow

Weakness:
* The novelty needs to be further justified
* More ablation studies would be ideal to better understand the model

**Summary Of The Paper:**

The authors propose a low-rank attention mechanism named DBA. Several experiments have been conducted to validate the proposed method's effectiveness.

**Summary Of The Review:**

My major concern is about the novelty of the proposed method. It would be great if the authors can make a table to compare the proposed method w/ other low-rank based attention modules on the their differences and why the contribution is not incremental. The authors do mention that the the proposed projection is not pre-determined, it would be better if the authors can better motivate why this pre-determination is not ideal and validate this from experiment results.

The authors list the hyper-parameters used in Sec. A.2, I wonder how these parameters may affect model performance. It would be better if the authors can conduct more ablation study to analyze the robustness of the proposed approach.

I also suggest the authors to describe how many seeds/repetitions each model has been trained in the experiments and report the accuracy variance.

---

> ### Author Response · Authors · 2022-11-18
> **Response to Reviewer tdpB (Part 1)**
>
> **Q1: The novelty needs to be further justified.**
>
> **A1:** Many thanks for your comments. The novelty of DBA lies in theory and model architecture perspectives. From the theory perspective, we first theoretically show that the sequence length can be compressed losslessly from a novel perspective of the information theory. In addition, we demonstrate that the hidden state dimension can be approximated by extending the Johnson–Lindenstrauss lemma with high-order small amount error.
>
> From the model architecture perspective, the sequence length compression matrices in DBA are dynamically determined by the input rather than in pre-determined or input invariant patterns like Linformer or Nyströmformer. Furthermore, DBA further compresses the hidden state dimension to achieve further efficiency.
>
> Due to the aforementioned differences, DBA has three main benefits over previous low-rank based approaches. Firstly, DBA could concentrate on different positions for different inputs to best preserve the most informative parts of a sequence, as analyzed in Section 1 and verified from the performance in Sections 4.2-4.4 and the visualization results in Section A.4. The DBA outperforms the Linformer over 5% in terms of average accuracy (62.29 verses 56.47) on LRA dataset and achieves higher efficiency in both speed and memory consumption perspectives. Secondly, DBA could process sequences in various lengths as the dimensions of sequence length compression matrices are also determined by the input. Thirdly, DBA could achieve state-of-the-art performance with high efficiency over various sequence length conditions due to jointly considering sequence length and hidden state dimension, as detailed illustrated in Sections 4.1 and 4.3. We modified Section 2.2 to better justify the superiority and novelty of DBA.

---

> > ### Author Response · Authors · 2022-11-18
> > **Response to Reviewer tdpB (Part2)**
> >
> > **Q2: More ablation studies would be ideal to better understand the model.**
> >
> > **A2:** Thanks for your contributing comment. We add ablation experiments of DBA regarding the efficiency and performance on the LRA dataset, as shown in Section A.5 and Tables 11-12. Results show that DBA could achieve a similar performance compared to the ablation counterparts without sequence length or hidden state dimension compression on three out of five tasks, which is consistent with the proposed theory that the sequence length compression is lossless and the hidden state dimension is approximated with high-order small amount error. The DBA achieves higher performance on the Image and Pathfinder datasets. We believe the principle of DBA contributes to the generalizability and ease of optimization, which achieves higher validation accuracy under the same training loss and faster coverage speed than the ablation counterparts, as shown in Figure 4. Note that DBA is faster and uses less memory compared to the ablation counterparts, demonstrating the efficiency and effectiveness of DBA.
> >
> > Different settings regarding the compression rates for sequence length and hidden state dimension are also illustrated in Tables 11-12. Results show that DBA could achieve balance between efficiency and performance with properly configured $d_p$ and $d_{in}$.
> >
> > Inference speed and average performance of DBA with different $d_p$.
> >
> > ||256 $\uparrow$|512 $\uparrow$|1k $\uparrow$|2k $\uparrow$|3k $\uparrow$|4k $\uparrow$|Avg. $\uparrow$|
> > |:-:|:-:|:-:|:-:|:-:|:-:|:-:|:-:|
> > |Vanilla Transformer|1.0|1.0|1.0|1.0|1.0|1.0|58.57|
> > |w/o optimize $n$|1.1|1.1|1.1|1.1|1.1|1.1|59.75±0.24|
> > |256|1.0|1.0|1.1|1.2|2.6|3.8|62.17±0.20|
> > |64|1.0|1.3|1.3|1.8|3.9|5.6|62.26±0.10|
> > |16|1.1|1.4|1.4|2.0|4.1|6.1|62.21±0.21|
> > |8|1.2|1.4|1.5|2.1|4.2|6.4|61.73±0.28|
> >
> > Peak memory consumption and average performance of DBA with different $d_p$.
> >
> > ||256 $\downarrow$|512 $\downarrow$|1k $\downarrow$|2k $\downarrow$|3k $\downarrow$|4k $\downarrow$|Avg. $\uparrow$|
> > |:-:|:-:|:-:|:-:|:-:|:-:|:-:|:-:|
> > |Vanilla Transformer|1.0|1.0|1.0|1.0|1.0|1.0|58.57|
> > |w/o optimize $n$|0.96|0.96|0.96|0.96|0.98|0.99|59.75±0.24|
> > |256|1.00|0.97|0.58|0.31|0.25|0.16|62.17±0.20|
> > |64|0.87|0.72|0.42|0.21|0.17|0.10|62.26±0.10|
> > |16|0.84|0.66|0.38|0.19|0.15|0.09|62.21±0.21|
> > |8|0.83|0.65|0.37|0.19|0.15|0.09|61.73±0.28|
> >
> > Inference speed and average performance of DBA with different $d_{in}$.
> >
> > ||256 $\uparrow$|512 $\uparrow$|1k $\uparrow$|2k $\uparrow$|3k $\uparrow$|4k $\uparrow$|Avg. $\uparrow$|
> > |:-:|:-:|:-:|:-:|:-:|:-:|:-:|:-:|
> > |Vanilla Transformer|1.0|1.0|1.0|1.0|1.0|1.0|58.57|
> > |w/o optimize $d$|1.0|1.1|1.4|1.9|4.0|5.9|60.95±0.32|
> > |64|1.1|1.4|1.4|1.9|4.1|5.9|62.26±0.16|
> > |24|1.1|1.4|1.4|2.0|4.1|6.1|62.21±0.21|
> > |12|1.2|1.4|1.4|2.0|4.2|6.3|61.37±0.18|
> >
> > Peak memory consumption and average performance of DBA with different $d_{in}$.
> >
> > ||256 $\downarrow$|512 $\downarrow$|1k $\downarrow$|2k $\downarrow$|3k $\downarrow$|4k $\downarrow$|Avg. $\uparrow$|
> > |:-:|:-:|:-:|:-:|:-:|:-:|:-:|:-:|
> > |Vanilla Transformer|1.0|1.0|1.0|1.0|1.0|1.0|58.57|
> > |w/o optimize $d$|0.85|0.68|0.39|0.20|0.16|0.09|60.95±0.32|
> > |64|0.84|0.67|0.38|0.20|0.16|0.09|62.26±0.16|
> > |24|0.84|0.66|0.38|0.19|0.15|0.09|62.21±0.21|
> > |12|0.84|0.66|0.38|0.19|0.15|0.09|61.37±0.18|
> >
> > Performance of DBA with different $d_p$ on the LRA benchmark.
> >
> > Models|ListOps $\uparrow$|Text $\uparrow$|Retrieval $\uparrow$|Image $\uparrow$|Pathfinder $\uparrow$|Avg. $\uparrow$|
> > |:-:|:-:|:-:|:-:|:-:|:-:|:-:|
> > w/o optimize $n$|38.08±0.23|66.00±0.10|78.53±0.04|41.76±0.54|74.37±0.28|59.75±0.24|
> > 256|37.53±0.08|66.20±0.14|80.81±0.14|48.03±0.17|78.30±0.46|62.17±0.20|
> > 64|37.40±0.05|66.17±0.12|80.69±0.09|46.98±0.09|80.05±0.17|62.26±0.10|
> > 16|38.10±0.40|66.25±0.04|80.64±0.01|46.51±0.50|79.56±0.10|62.21±0.21|
> > 8|37.88±0.33|66.06±0.13|80.37±0.19|45.70±0.30|78.65±0.46|61.73±0.28|
> >
> > Performance of DBA with different $d_{in}$ on the LRA benchmark.
> >
> > Models|ListOps $\uparrow$|Text $\uparrow$|Retrieval $\uparrow$|Image $\uparrow$|Pathfinder $\uparrow$|Avg. $\uparrow$|
> > |:-:|:-:|:-:|:-:|:-:|:-:|:-:|
> > w/o optimize $d$|37.55±0.25|65.92±0.10|79.98±0.19|45.39±0.34|75.90±0.73|60.95±0.32|
> > 64|37.50±0.05|66.19±0.09|80.22±0.13|47.64±0.16|79.76±0.35|62.26±0.16|
> > 24|38.10±0.40|66.25±0.04|80.64±0.01|46.51±0.50|79.56±0.10|62.21±0.21|
> > 12|37.65±0.15|66.18±0.10|78.14±0.13|46.24±0.36|78.65±0.18|61.37±0.18|
> >
> > Furthermore, we visualized the dynamic sequence lengths compression matrices in DBA and compared them with the input invariant compression matrix in Linformer, as shown in Section A.4 and Figure 3. Results show that the DBA sequence length projection matrix is determined by the input, which highlights values in different positions for the different inputs, while the sequence length projection matrix in Linformer concentrates on the same position for different samples.

---

> > > ### Author Response · Authors · 2022-11-18
> > > **Response to Reviewer tdpB (Part 3)**
> > >
> > > **Q3: Overall the technical details are sufficient and the experiment results look promising. It would be better if the authors can further justify the novelty of the proposed method.**
> > >
> > > **A3:** Thanks for your comment. The novelty of DBA is from theory and model architecture perspectives. From the theory perspective, we first theoretically show that the sequence length can be compressed losslessly from a novel perspective of the information theory. In addition, we demonstrate that the hidden state dimension can be approximated by extending the Johnson–Lindenstrauss lemma with high-order small amount error.
> > >
> > > From the perspective of model architecture, the sequence length compression matrices in DBA are dynamically determined by the input rather than in pre-determined or input invariant patterns like Linformer or Nyströmformer. Furthermore, the DBA further compresses the hidden state dimension to achieve further efficiency.
> > >
> > > Due to the aforementioned differences, DBA has three main benefits over previous low-rank based approaches. Firstly, DBA could concentrate on different positions for different inputs to best preserve the most informative parts of a sequence, as analyzed in Section 1 and verified from the performance in Sections 4.2-4.4 and the visualization results in Section A.4. The DBA outperforms the Linformer over 5% in terms of average accuracy (62.29 verses 56.47) on LRA dataset and achieves higher efficiency in both speed and memory consumption perspectives. Secondly, DBA could process sequences in various lengths as the dimensions of sequence length compression matrices are also determined by the input. Thirdly, DBA could achieve state-of-the-art performance with high efficiency over various sequence length conditions due to jointly considering the sequence length and hidden state dimension, as detailed illustrated in Sections 4.1 and 4.3. We modified Section 2.2 to better justify the superiority and novelty of DBA.

---

> > > > ### Author Response · Authors · 2022-11-18
> > > > **Response to Reviewer tdpB (Part 4)**
> > > >
> > > > **Q4: My major concern is about the novelty of the proposed method. It would be great if the authors can make a table to compare the proposed method w/ other low-rank based attention modules on the their differences and why the contribution is not incremental. The authors do mention that the the proposed projection is not pre-determined, it would be better if the authors can better motivate why this pre-determination is not ideal and validate this from experiment results.**
> > > >
> > > > **A4:** Thanks for your contributing comment. We highlight the key differences between DBA and other low-rank based attention models, including Linformer and Nyströmformer, as shown in the following Table. The DBA compresses sequence length in various lengths with dynamic projection matrices determined by input for all $Q$, $K$, $V$ in Transformer and also focuses on the impact of hidden state dimension, while the previous models compress sequence length with input invariant patterns and ignored the impact of hidden state dimension.
> > > >
> > > > **Comparison of DBA with the previous low-rank based attention approaches.**
> > > > |Model|Sequence Length Compression Matrix|Process Sequence in Various Length|Compressed Variables in Transformer|Hidden State Dimension Compression|Applied Models|Average Performance $\uparrow$|
> > > > |:-:|:-:|:-:|:-:|:-:|:-:|:-:|
> > > > |Linformer|Learned parameters (input invarient)|-|$K$, $V$|-|Transformer|56.47|
> > > > |Nyströmformer|Mean pooling (pre-determined)|-|$Q$, $K$|-|Transformer|58.95|
> > > > |**DBA**|Dynamically determined by the input|$ \checkmark$|$Q$, $K$, $V$|$ \checkmark$|Transformer, State Space Model|**62.21±0.21**|
> > > >
> > > > Due to the aforementioned differences, DBA has three major benefits compared to previous low-rank based methods. Firstly, DBA could concentrate on different positions for different inputs to best preserve the most informative parts of a sequence, which is analyzed in Section 1 and validated from the performance in Sections 4.2-4.4 and visualization results in Section A.4. Secondly, DBA could process sequences in various length as the dimensions of sequence length compression matrices are also determined by the input. Thirdly, DBA could achieve state-of-the-art performance with high efficiency over various sequence length conditions due to jointly considering the sequence length and hidden state dimension.
> > > >
> > > > As for the second concern about why the input invariant projection is not ideal, we illustrate it from theoretical and experimental perspectives. From the theoretical perspective, the informative part of a sequence varies from sequence to sequence. Hence, adopting input invariant sequence length compression might fail to preserve the most informative parts lying in different positions and limit the performance over tasks where the most informative parts of inputs change.
> > > >
> > > > From the experimental perspective, we address it in three different aspects. Firstly, we visualized the dynamic sequence lengths compression matrices in DBA and compared them with the input invariant compression matrix in Linformer on the Selfregulationscp1 task, as shown in Section A.4 and Figure 3. The Selfregulationscp1 record the EEG data and is one of the UEA multivariate time series classification archives. Results show that the sequence length projection matrix in DBA is determined by the input, which highlights values in different positions for the different inputs, while Linformer concentrates on the same position for different samples. Secondly, the sequence compression matrices in DBA is “smoother” between adjacent positions and have more noticeable trends than the compression matrix in Linformer. From the characteristics of the Selfregulationscp1 task and the way people diagnose such diseases, the concentrated position shall be different for different inputs and share more coherent trends between adjacent points like in DBA rather than oscillate. The DBA process the signal more human-like and could achieve higher performance. Finally, the proposed DBA outperforms the previous low-rank based attention models using input invariant sequence length compression matrices and achieves higher efficiency, as shown in Section 4.2-4.4 and Tables 2-4. Based on the aforementioned theoretical and experimental aspects, we believe that the dynamic sequence length compression matrices in DBA are better than the input invariant counterparts.

---

> > > > > ### Author Response · Authors · 2022-11-18
> > > > > **Response to Reviewer tdpB (Part 5)**
> > > > >
> > > > > **Q5: The authors list the hyper-parameters used in Sec. A.2, I wonder how these parameters may affect model performance. It would be better if the authors can conduct more ablation study to analyze the robustness of the proposed approach.**
> > > > >
> > > > > **A5:** Thanks for your comment. We conduct ablation studies regarding the sequence length compression hyper-parameter $d_p$ and hidden state dimension compression hyper-parameter $d_{in}$ to the DBA, as shown in Section A.5 and Tables 11-12. Results show that with $d_p$ in 8-256 and $d_{in}$ in 12-64, the final performances are generally stable (around 1% fluctuation in terms of average accuracy), while the speed and peak memory usage have great changes. In conclusion, the DBA could achieve high performance and efficiency simultaneously with properly configured hyper-parameters $d_p$ and $d_{in}$,.
> > > > >
> > > > > **Q6: I also suggest the authors to describe how many seeds/repetitions each model has been trained in the experiments and report the accuracy variance.**
> > > > >
> > > > > **A6:** Thanks for your comment. We trained the DBA on each task with 5 random seeds and report the accuracy variance of DBA, as shown in Tables 2-4 and 12.

---

> ### Author Response · Authors · 2022-11-27
> **Response to Reviewer tdpB**
>
> Dear reviewer, we have tried to address your constructive comments in the earlier response. Please feel free to contact us if there are any further questions or suggestions. Discussions are always welcome.

---

> ### Author Response · Authors · 2022-11-30
> **Response to Reviewer tdpB**
>
> Dear reviewer,
>
> Thank you for your valuable comments and suggestions in reviewing our manuscript. The constructive comments you addressed have enabled us to disseminate our manuscript to the higher possible quality, and we have made extensive revisions according. Please let us know if there are any further comments or suggestions. Just as a reminder, the discussion stage is due by Dec 12, 2022.
>
> We are really grateful for your thorough review and looking forward to hearing from you in due course.

---

> ### Author Response · Authors · 2022-12-11
> **Reminder for Reviewer tdpB**
>
> Dear reviewer,
>
> Thank you for your constructive comments and suggestions in reviewing our manuscript. We have made extensive revisions according to your valuable comments point by point. Please let us know if there are any further suggestions or comments. Just as a reminder, the discussion stage is due by Dec 12, 2022.
>
> We are really grateful for your thorough review and are looking forward to hearing from you as soon as possible.

---

### Author Response · Authors · 2022-11-18
**General Response to All Reviewers**

We would like to express our appreciation to all reviewers for the valuable and constructive comments to our manuscript.
Overall, we are encouraged that they find that:
1.  The learnable projection in the paper is novel and squeezes the computation not only in terms of sequence length but also in terms of hidden dimension. (Reviewer PZVd)
2.  The results of DBA are impressive compared to other linear attention-based models. (Reviewer 9GWA, PZVd)
3.  The technical details are presented clearly and with details. (Reviewer tdpB)

We have carefully studied all the comments and have made extensive revisions accordingly. The main revisions of our manuscript are as follows:
1.  We performed more ablation experiments of DBA and visualized the dynamic sequence length compression matrices to better illustrate the characteristics of DBA (Sections A.4, A.5, Tables 11, 12, and Figures 3, 4).
2.  We extend DBA to the state space model (i.e., S4), and the results show that the S4 with DBA optimization could achieve 1.4x average speed-up and 0.8x average memory usage with competitive performance compared to the baseline, demonstrating the universality of DBA (Sections 4.5, A.3 and Table 5).

Next, we address each reviewer's detailed concerns point by point. We hope we have addressed all of your concerns. Discussions are always open. Thank you!

---

### Decision · Program_Chairs · 2023-01-20

**Decision:**

Reject

**Justification For Why Not Higher Score:**

The main concern is writing and presentation. You can not write equality and claim information theory proof for something that does not hold (exchanging matrix multiplication with softmax). I think, this has to be completely rewritten.

**Justification For Why Not Lower Score:**

N/A

**Metareview: Summary, Strengths And Weaknesses:**

The authors replace the attention mechanism with DBA (dynamic bilinear attention). It replaces input sequences with shorter ones and also projects queries and key vectors to lower dimensionalities. The projection matrix is computed using cross-attention type layer, which is potentially interesting.  The numerical experiments show that this architecture is quite competive.


Weaknesses: The paper is not well-written. The algorithm is not summarized anywhere, so the reader has to guess how the layers is actually implemented (i.e. how Q_DBA is computed? how project matrix is computed? What are learnable parameters?)
The derivation of the formulas in Section 3.1 has no mathematical meaning (i.e. the notation basis_r, basis_c of a matrix), as well as "can be represented losslessly", but then the formula (6) follows ("in practice we could form"), which is just the compression of a low-rank matrix (makes sense), but it is not a derivation, and it has no connection to information theory. Basically, formula (8) is a different formula, can be used as a starting point, but please, don't call this derivation from a theory. These are just completely different formulas, you can not simply exchange softmax & matrix multplication. The formula (9) has a standard (awful) 1/eps^2 dependence on the error, so again you can use this as a "proof".  For the typical sizes, the error of approximation will be more than 100%. As a motivation it is ok, but all the text has equality (not even a single approximately equal sign).



**Summary Of Ac-Reviewer Meeting:**

N?A